# Identifying Latent State-Transition Processes for Individualized Reinforcement Learning

**Yuewen Sun**[1,2]**, Biwei Huang**[3]**, Yu Yao**[4]**, Donghuo Zeng**[5]**, Xinshuai Dong**[2]**, Songyao Jin**[3]**,
**Boyang Sun**[1]**, Roberto Legaspi**[5]**, Kazushi Ikeda**[5]**, Peter Spirtes**[2]**, Kun Zhang**[1,2]

[1]Mohamed bin Zayed University of Artificial Intelligence, [2]Carnegie Mellon University,
[3]University of California San Diego, [4]The University of Sydney, [5]KDDI Research

## Abstract

The application of reinforcement learning (RL) involving interactions with individuals has grown significantly in recent years. These interactions, influenced by factors such as personal preferences and physiological differences, causally influence state transitions, ranging from health conditions in healthcare to learning progress in education. As a result, different individuals may exhibit different state-transition processes. Understanding individualized state-transition processes is essential for optimizing individualized policies. In practice, however, identifying these state-transition processes is challenging, as individual-specific factors often remain latent. In this paper, we establish the identifiability of these latent factors and introduce a practical method that effectively learns these processes from observed state-action trajectories. Experiments on various datasets show that the proposed method can effectively identify latent state-transition processes and facilitate the learning of individualized RL policies.

## 1 Introduction

Reinforcement Learning (RL) [44] involves training agents to make decisions by interacting with the environment. The agent observes its current state, takes an action, and transitions to a new state with a reward. Such a sequence of moving from one state to another is known as a *state-transition process*.

Individualized RL focuses on adapting the policy for each individual. It has recently seen increasing application in various sectors, including healthcare [86, 55, 19], education [66, 2, 12], and e-commerce [51, 84, 1]. The *individual-specific factors* [58], which capture the unique characteristics of each individual, play an important role in causally influencing the transitions between states. These factors cover a range of aspects, including individual preferences, past experiences, and physiological differences. For example, in education, different learning styles can influence how students with the same prior knowledge benefit from a tutorial. In healthcare, genetic differences can affect how patients with hypertension respond to the same treatments. Understanding individual-specific factors is essential for designing better RL systems that provide more individualized and effective decisions [39, 26, 3, 63]. By understanding learning styles in education, RL agents can recommend personalized tutorials, such as animated content for visual learners or hands-on exercises for kinesthetic learners. Similarly, in healthcare, knowledge of genetic makeup can help agents suggest treatment plans tailored to individual needs, leading to improved health outcomes.

However, these individual-specific factors are not always observable, making it challenging to understand the Latent Individualized State-Transition (LIST) processes, as illustrated in Figure 1(a). These latent factors, such as learning styles and genetic makeup, are unique to each individual and have a time-invariant influence on the state-transition process. This raises the question: can the identifiability of these latent factors be guaranteed?

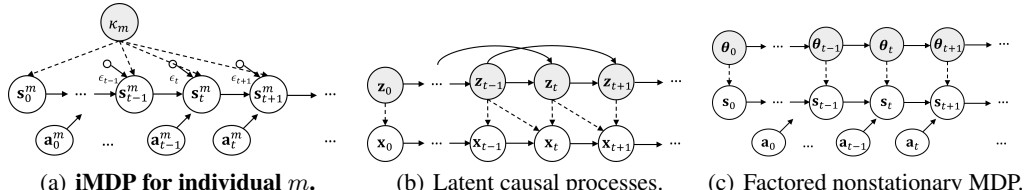

(a) **iMDP for individual** $m$.  (b) Latent causal processes.  (c) Factored nonstationary MDP.

Figure 1: Comparisons of different state-transition processes. Latent variables are colored in grey.

Such identifiability is easier to achieve when the observations are either i.i.d., or i.i.d. given side information (e.g., domain index, time index, etc.), by exploiting sparsity [94], variability [45], or functional complexity [41]. To the best of our knowledge, only a few studies have explored the identifiability of latent factors from temporal observations. These methods primarily focus on the time-varying latent factors, which differs from our work on time-invariant latent factors. Specifically, existing works [83, 82, 6] assume time-varying latent variables without considering the influence of actions in the generative process (see Figure 1(b)). Factored MDPs [13] incorporate actions into the process but still assume that the latent factors change over time (see Figure 1(c)). Thus, the findings from these studies cannot be applied to our setting. Intuitively, this is because time-invariant latent factors cannot provide the variability that many current methods rely on to achieve identifiability. Therefore, it remains unclear how to derive the identifiability of the latent individual-specific factors and the corresponding latent state-transition processes from observed states and actions.

Recent advances in finite mixture models [74, 65] have demonstrated strong identifiability results by exploiting group information in nonparametric settings. By assuming that observations within the same group are known to come from the same component, the mixture of probability measures can be uniquely identified under appropriate assumptions. Inspired by these works, we establish the identifiability of latent factors by leveraging group information from the data, making it easier to distinguish different underlying components. We propose both a finite latent, nonparametric setting and an infinite latent, parametric setting, and develop a theoretically grounded framework that effectively learns these processes from observed state-action trajectories. Our contributions are summarized as follows:

- We introduce Individualized Markov Decision Processes (iMDPs), a novel framework that incorporates latent individual-specific factors $\kappa$ into state-transition processes. We treat such latent factors that influence each state in the decision process and vary across individuals.
- Our work provides theoretical guarantees and new insights into learning state-transition processes with latent factors. For finite $\kappa$, we consider two scenarios to establish identifiability in nonparametric settings. For infinite $\kappa$, we demonstrate identifiability in the post-nonlinear case. To the best of our knowledge, this is the first work to provide a theoretical guarantee for identifying latent individual-specific factors from observed transitions.
- We propose a practical generative method that can effectively estimate the latent individual-specific factors. Empirical results on various datasets demonstrate the effectiveness of the method both in inferring latent factors and in learning individualized policies.

## 2   Related Work

**Individualized Machine-Learning Applications**   Recently, machine learning has created highly individualized solutions across various domains. In healthcare, algorithms support individualized treatment for physical activity, weight loss, and diabetes management [86, 55, 16, 15]. In finance, it provides accurate stock predictions for stock market activities [54]. Education is benefiting from individualized ICT systems that address the individual learning needs of students [14, 38]. Furthermore, transportation has seen the development of individualized car-following strategies [67] that improve driving safety and efficiency. Meanwhile, entertainment platforms such as YouTube and TikTok are using it to provide individualized video recommendations [5, 31].

**Reinforcement Learning with Latent State-Transition Processes**   In the field of RL, various models explore the state transition dynamics with latent variables. One such approach is Partially Observable Markov Decision Processes (POMDPs) [62], where the full information about the state is

unknown. In POMDPs, observations are generated from the latent states, which do not match our individual latent setting. For example, block MDPs [89, 93] assume that there is a fixed and unknown mapping from observations to the latent states. Factored MDPs [33, 13], which provide the partial identifiability of latent factors, assume that the latent factors evolve over time following a Markov process. On the other hand, there exists a piece of work focusing on estimating state transitions with time-invariant latent factors. Models such as contextual MDPs [27, 58, 63], latent MDPs [49, 48] and multitask RL [71, 22] consider similar scenarios with our latent individual-specific factors. However, these works lack theoretical guarantees on the identifiability of the latent factors thus it is hard for them to guarantee individualized decision-making.

## 3 Problem Formulation

Consider a population with $M$ individuals that can be divided into $G$ groups, where the exact group memberships are unknown. We introduce iMDPs to model individualized decision-making processes, where *observed* individual uniqueness is captured by $u$, and *latent* group-level properties are encoded by $\kappa$. Specifically, each individual is assigned a unique value of $u$, with the cardinality of $u$ being $M$. Meanwhile, individuals within the same group share the same value of $\kappa$, which differs across groups, and the cardinality of $\kappa$ is $G$. For each individual, the value of $\kappa$ is predetermined and $\kappa$ has a time-invariant influence on the state-transition process. Furthermore, all individuals are assumed to share the same state and action spaces. The iMDP is defined as follows.

**Definition 3.1** (iMDP). *An iMDP consists of a tuple $\langle \mathcal{S}, \mathcal{A}, R, \{s_0^m\}_{m=1}^M, \{u_m\}_{m=1}^M, \{\mathbb{T}_m\}_{m=1}^M \rangle$, where $M$ is the number of individuals; $\mathcal{S}$ and $\mathcal{A}$ are the state and action spaces, respectively; $R \in \mathbb{R}$ is the immediate reward received after transitioning from current state $s$ to new state $s'$ via action $a$, i.e., $r = R_a(s, s')$ for $s \in \mathcal{S}$, $s' \in \mathcal{S}$ and $a \in \mathcal{A}$. To model individualized decision-making, each individual is associated with a unique index and an individualized state-transition process. In the $m^{th}$ MDP, $u_m$ is the unique index identifying each individual, and $s_0^m$ is the individualized initial state. The individualized state-transition probability is denoted by $\mathbb{T}_m := \mathbb{P}_m(s'|s, a, \kappa_m) \in \mathbb{R}^{|\mathcal{S}| \times |\mathcal{A}| \times |\mathcal{S}|}$, where $\kappa_m$ is the latent individual-specific factor with a cardinality of $G$. Thus, the joint distribution of any adjacent state-action pairs $(s, a, s')$ can be fully characterized by $u$ and $\kappa$ as:*

$$\mathbb{P}(s, a, s'|u) = \mathbb{P}(s'|s, a, \kappa)\mathbb{P}(s, a|u). \tag{1}$$

**Data Generation Process** Here we introduce the LIST processes based on iMDP framework. For individual $m$, the observed states $\mathbf{s}_t^m$ are generated according to the following generation process:

$$\mathbf{s}_t^m = f(\mathbf{s}_{t-1}^m, \mathbf{a}_{t-1}^m, \kappa_m, \epsilon_t^m), \tag{2}$$

where $\mathbf{s}_t^m = (s_{0,t}^m, \ldots, s_{d_s,t}^m)^\top \in \mathbb{R}^{d_s}$ denotes the $d_s$-dimensional state at time $t$, and $\mathbf{a}_t^m = (a_{0,t}^m, \ldots, a_{d_a,t}^m)^\top \in \mathbb{R}^{d_a}$ denotes the $d_a$-dimensional action at time $t$. The term $\epsilon_t^m$ represents independent noise, while $\kappa$ characterizes the group-level properties within the population. The transition function $f$ is identical across individuals, governs the dynamics and is consistent with Eq. (1). During interaction with the environment, the trajectory $\tau_m = \{\mathbf{s}_0^m, \mathbf{a}_0^m, \mathbf{s}_1^m, \ldots, \mathbf{s}_T^m\}$ is recorded as a sequence of observed state-action tuples, where $T$ denotes the length of the trajectory.

**Objectives** In this work, we investigate RL with a focus on capturing individualized latent state-transition dynamics. Our objectives are twofold: 1) to identify the latent individual-specific factors $\kappa$ from observed trajectories, and 2) to learn individualized policies for each agent, facilitating policy adaptation for newcomers. Consider the example of hypertension diagnosis in healthcare. Treating all patients identically may lead to different outcomes due to the dynamics of state transitions, which are influenced by latent $\kappa$. Therefore, accurate identification of $\kappa$ from the population provides crucial dynamic insights. Once $\kappa$ is uncovered, we can categorize patients into different groups and tailor individualized treatments for each, which motivates our second goal.

## 4 Identifiability Analysis

We establish the identifiability of latent individualized state-transition processes under two conditions: (1) finite latent condition under group determinacy assumption (see Theorem 4.1 and Theorem 4.2) and (2) infinite latent condition under functional constraint (see Theorem 4.3). The corresponding identifiability results are presented below.

**Finite Latent Condition**   Suppose the value of $\kappa$ is finite; we first provide the definition of group-wise identifiability. For the detailed assumptions discussion and proofs, please see Appendix B.

**Definition 4.1** (Group-wise Identifiability). *Let $\{\tau_m\}_{m=1}^M$ be sequences of observed states and actions collected from $G$ groups under a fixed policy, following the true latent individualized state-transition processes described in Eq. (2). A learned generative model $(\hat{f}, \hat{\kappa}, \hat{\epsilon})$ is observational equivalent to $(f, \kappa, \epsilon)$ if the joint distribution $\mathbb{P}_{\hat{f}, \hat{\kappa}, \hat{\epsilon}}(s, a, s')$ matches $\mathbb{P}_{f, \kappa, \epsilon}(s, a, s')$ everywhere. We say that the latent individualized state-transition processes are group-wise identifiable if observational equivalence can always lead to the identifiability of latent individual-specific factors across the population up to the invertible transformation $g$:*

$$\mathbb{P}_{\hat{f}, \hat{\kappa}, \hat{\epsilon}}(s, a, s') = \mathbb{P}_{f, \kappa, \epsilon}(s, a, s') \iff \hat{\kappa} = g(\kappa). \tag{3}$$

**Assumption 4.1** (Group Determinacy). *Consider a finite mixture model $\sum_{g=1}^G \pi_g \delta_{\kappa_g}(\kappa)$, where $\pi_g$ represents mixing proportions with $\sum_{g=1}^G \pi_g = 1$, and $\delta_{\kappa_g}$ is the Dirac function centered at $\kappa_g$. Each unique value of $\kappa$ corresponds to a specific group in the population, with $\delta_{\kappa_g}(\kappa) = 1$ if $\kappa = \kappa_g$ and 0 otherwise, and the number of individuals per group is greater than $2G - 1$.*

Assumption 4.1 guarantees that identifiability can be derived from the finite mixture model perspective using group information. We consider two scenarios to establish identifiability under finite latent conditions. Theorem 4.1 considers finite samples, establishing identifiability under specific assumptions about the initial states $\{s_0^m\}_{m=1}^M$. Theorem 4.2 guarantees asymptotic identifiability for sufficiently long trajectories without imposing constraints on the initial states.

**Theorem 4.1.** *Assume the LIST processes described in Eq. (2). Suppose the distributions of initial states are the same for all individuals within the same groups, and the trajectory length is finite. Under Assumption 4.1, the identifiability of the individual-specific factor $\kappa$ is guaranteed.*

**Theorem 4.2.** *Assume the LIST processes described in Eq. (2). Suppose the distribution of the initial state varies across individuals, and the trajectory length is sufficiently long, i.e., there exist two different individuals within the same group have overlap condition $\mathbb{P}(s, a|u = u_i) = \mathbb{P}(s, a|u = u_j)$, where $i \neq j$. Under Assumption 4.1, the identifiability of $\kappa$ is asymptotically guaranteed.*

**Infinite Latent Condition**   Theorem 4.3 demonstrates that under specific functional constraints, the identifiability of latent individual-specific variables can be extended to multiple and infinite latent factors. Specifically, we consider the post-nonlinear temporal model [90] and allow multiple instances of $\kappa$ to influence the state-transition dynamics. The identifiability and cardinality of latent factors are decided by the rank conditions of specific covariance submatrices derived from the observed data. Furthermore, empirical results in Section 6 indicate that even when multiple latent factors with infinite cardinality are present, our estimation framework (see Section 5) still encourages the identification of latent factors. For a detailed proof, please see Appendix C.

**Theorem 4.3.** *Consider a trajectory collected from the post-nonlinear temporal model (Definition C.1) with $d_s$-dimensional observed states over time $t = 1, \ldots, T$. Let $m$ latent factors $\kappa_j$, $j = 1, \ldots, m$, have direct causal influence on all states, and $\mathcal{S}_t = \{s_{1,t}, s_{2,t}, \ldots, s_{d_s,t}\}$ represent the set of all state variables at time $t$. These latent factors, as well as the state-transition process, can be identified if and only if for every $i = m + 2, \ldots, T - (m + 1)$, there exist pairs of minimal rank sets (Definition C.2) $(\mathbf{A_i}, \mathbf{B_i})$, denoted as $\mathbf{A_i} = \mathcal{R}_{i,i^-}$ and $\mathbf{B_i} = \mathcal{R}_{i,i^+}$, where $i^- < i < i^+$, that satisfy:*

- *(Rank Deficiency for Identification) In addition to $\mathcal{S}_i$ shared by $\mathbf{A_i}$ and $\mathbf{B_i}$, each subset should include a randomly selected set of $m + 1$ additional state variables. If the covariance matrices $\Sigma_{\mathbf{A_i}, \mathbf{B_i}}$ exhibit a consistent rank $r$ (where $r > d_s$) across all distinct indices $i$, this consistency implies the existence of latent factors within the system.*

- *(Quantification of Latent Factors) Once identification is established, the number of latent factors $m$ can be inferred from the rank deficiency of $\Sigma_{\mathbf{A_i}, \mathbf{B_i}}$ relative to the dimensionality of the observed variables, specifically given by $m = \mathrm{rank}(\Sigma_{\mathbf{A_i}, \mathbf{B_i}}) - d_s$.*

## 5   Estimation and Policy Learning Framework

**Overview**   We propose a two-stage approach that addresses two key objectives: (1) developing an estimation framework to recover the latent factors $\kappa$ from individual trajectories, and (2) implementing

individualized policy learning to facilitate policy adaptation for new individuals. The estimation framework is designed to meet the conditions outlined in the identifiability theorems. According to Definition 4.1, identifiability is achieved if and only if observational equivalence implies latent factor equivalence. This motivates the use of a generative model to estimate latent factors and ensure that the reconstructed distribution aligns closely with the true observed distribution.

To fulfill this requirement, we make a modification to the variational autoencoder (VAE) architecture [43], which enables the unsupervised estimation of latent factors. Theorems 4.1 and 4.2 guarantee the asymptotic accuracy of this alignment. As illustrated in Figure 2, individual trajectories are encoded in a discrete embedding space, which is consistent with the assumptions in the theorems. Detailed pseudocode for the proposed approach is provided in Appendix I, and a comprehensive description of each component is provided in Appendix H.

## 5.1 Latent Estimation Framework

**Temporal Encoding and Latent Factor Quantization**   The group determinacy assumption suggests the identifiability of the latent individual-specific factor $\kappa$. Given that $\kappa$ is time-invariant and influences each state in the transition process, we begin by employing a sequential encoder to capture the high-level representation, denoted as $z_m$, based on the input from all states across each trajectory. We then utilize a vector quantization layer [73] to discretize the latent space and estimate the latent factor $\hat{\kappa}_m$. This quantization step ensures that the learned representation aligns with the group-level characteristics of the latent factors, as assumed in our framework, thereby supporting our objectives.

Specifically, to capture temporal dependency from sequential observations, we use sequential neural networks such as Conv1D [50] or Long Short-Term Memory (LSTM) [29] as encoders. The encoder function, denoted as $g$, maps the input trajectory $\{\mathbf{s}_0^m, \ldots, \mathbf{s}_T^m\}$ to a continuous high-level representation $z_m = g(\mathbf{s}_0^m, \ldots, \mathbf{s}_T^m)$, where Conv1D extracts local temporal patterns from subsequences, and LSTM aggregates information over time by sequentially updating its hidden and cell states. After processing the whole trajectory sequentially, the final hidden state of the LSTM and the final output of the Conv1D layer serve as the representation $z_m$.

However, continuous representations $z_m$ are incompatible with our framework's requirements. To address this, we apply a vector quantization layer to discretize the latent space and approximate the latent factor. This layer maps $z_m$ to the nearest vector in a predefined embedding dictionary $E = \{e_1, e_2, \ldots, e_G\}$, where each vector $e_i$ represents a distinct group in the discrete embedding space. The assignment is realized by finding the nearest neighbor in the dictionary as $\hat{\kappa}_m = \arg\min_{e_i} \|z_m - e_i\|_2$, where $\hat{\kappa}_m$ represents the quantized vector $e_i$ closest to the continuous representation $z_m$.

**Latent Factor Estimation via Conditional Reconstruction**   Reconstruction plays a critical role in unsupervised latent factor estimation by ensuring that the reconstructed distribution closely matches the true observed distribution. As stated in Definition 4.1, this alignment allows the estimated latent factors to approximate the true latent factors. Given the nature of the transition processes, we design a conditional decoder that uses the state-action pairs $(\mathbf{s}_{t-1}^m, \mathbf{a}_{t-1}^m)$ as conditions to guide the reconstruction of $\hat{\mathbf{s}}_t^m$. These conditions, together with the estimated latent factors $\hat{\kappa}_m$, serve as inputs to the decoder. The reconstruction accuracy is evaluated through the reconstruction likelihood $p_{\text{Recon}}(\hat{\mathbf{s}}_t^m | \mathbf{s}_{t-1}^m, \mathbf{a}_{t-1}^m, \hat{\kappa}_m)$, where $p_{\text{Recon}}$ denotes the reconstruction distribution. It provides a probabilistic measure of how well $\hat{\mathbf{s}}_t^m$ approximates $\mathbf{s}_t^m$. This likelihood quantitatively evaluates the performance of the decoder and the ability of the model to accurately reconstruct observed states.

**Training Objectives**   The parameters are optimized according to the following ELBO objective:

$$\mathcal{L}_{\text{ELBO}} = \mathcal{L}_{\text{Recon}} + \alpha \mathcal{L}_{\text{Quant}} + \beta \mathcal{L}_{\text{Commit}} \tag{4}$$

where $\alpha$ and $\beta$ are weights for the corresponding loss components. Specifically, (1) *Reconstruction loss* $\mathcal{L}_{\text{Recon}} = \sum_t \|\mathbf{s}_t^m - \text{De}(\text{En}(\mathbf{s}_{0:T}^m), \mathbf{s}_{t-1}^m, \mathbf{a}_{t-1}^m)\|^2$. This measures the discrepancy between the reconstructed state $\hat{\mathbf{s}}_t^m$ and the original state $\mathbf{s}_t^m$, where En and De are the encoder and decoder, respectively. (2) *Quantization loss* $\mathcal{L}_{\text{Quant}} = \sum_i \|\text{sg}[z_{m,i}] - e_{m,i}\|^2$. This evaluates the discrepancy between the encoder output $z_m$ and its discretized representation $e_m$. Since the quantization step is undifferentiable, we use the stop-gradient operation $\text{sg}[\cdot]$ to update the dictionary vectors without affecting the encoder parameters. (3) *Commitment loss* $\mathcal{L}_{\text{Commit}} = \sum_i \|z_{m,i} - \text{sg}[e_{m,i}]\|^2$. This term minimizes the discrepancy between $z_m$ and $e_m$, encouraging $z_m$ to align closely with the embedding

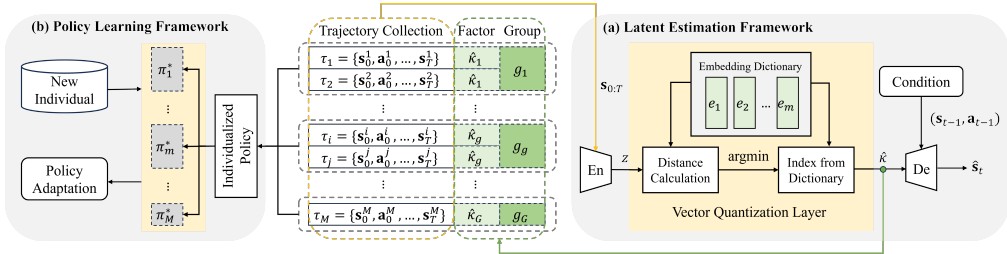

Figure 2: (a) Latent estimation framework takes each trajectory $\mathbf{s}_{0:T}$ as input and processes it through a quantized encoder to estimate the latent factor $\hat{\kappa}$. A conditional decoder then reconstructs $\hat{\mathbf{s}}_t$, using $(\mathbf{s}_{t-1}, \mathbf{a}_{t-1})$ as condition and $\hat{\kappa}$ as input. (b) Once the estimated latent factors are assigned to each trajectory, the policy learning framework integrates these latents as augmented labels to optimize the RL policy. For new individuals, the initial policy is adapted based on their group affiliation, enabling individualized policy adaptation for newcomers.

space. By applying the stop gradient to $e_{m,i}$, gradients from this loss do not change the dictionary vectors, but instead optimize the encoder parameters.

## 5.2 Policy Learning Framework

The estimation network is pre-trained offline. When a new individual arrives, we estimate $\kappa$ and adapt the policy simultaneously through new interactions. Specifically,

**Latent-based Policy Individualization** The estimated latent individual-specific factors $\hat{\kappa}$, together with the offline trajectories from all individuals, are used to learn the individualized policy $\pi_\kappa^*$. We view the estimated factors as an augmented component of the policy input, and the policy training objective is adjusted to match the unique characteristics of each individual.

Take Q-learning [56] as an example. In the individualized process, the latent factor is augmented as a policy input as $\mu_\pi(\mathbf{s}_t; \theta^\mu) \to \mu_\pi^m(\mathbf{s}_t^m, \hat{\kappa}_m; \theta^\mu)$, where $\theta^\mu$ represents the parameters of the policy network. This incorporation allows the policy to adapt effectively to individual-specific dynamics. The training objective is updated accordingly as $\mathcal{J}(\theta^\mu) = \mathbb{E}\left[\sum_{t=0}^\infty \gamma^t Q\left(\mathbf{s}_t, \mu_\pi^m(\mathbf{s}_t^m, \hat{\kappa}_m; \theta^\mu); \theta^Q\right)\right]$, where $\gamma$ is the discount factor, and $Q$ represents the Q value. Such individualization improves policy adaptability across varying environments, and our framework is general enough to be seamlessly integrated with various RL algorithms.

**Policy Adaptation for New Individual** Policy adaptation for a new individual involves two steps: initializing the policy from the individualized policy $\pi_\kappa^*$ and fine-tuning it through new interactions. For a new individual from group $g_n$, the group factor $\hat{\kappa}_n$ is first estimated. The policy network $\pi_{\text{new}}$ is then initialized by directly transferring parameters from $\pi_{\kappa=\hat{\kappa}_n}^*$. Specifically, $\pi_{\text{new}} = \arg\max_\pi \mathbb{E}_{(\mathbf{s}_t, \mathbf{a}_t) \in D_n}(R_{\mathbf{a}_t}(\mathbf{s}_t))$ is fine-tuned based on the new individual's trajectory $D_n$ by maximizing the expected reward. The dataset $D_n$ is incrementally augmented with new observations $(\mathbf{s}_t, \mathbf{a}_t, R_t, \mathbf{s}_{t+1})$ collected under $\pi_{\text{new}}$. This process refines the policy to better fit the specific characteristics of the new individual. Additionally, the updated $D_n$ helps to accurately estimate the latent factor of the new individual, further improving the adaptation process.

## 6 Experiment

**Evaluation Metrics** To measure the latent identification, we quantify the correlation between the estimated and true latent factors using: (1) Pearson Correlation Coefficient (PCC) for single latent, which quantifies the linear correlation between the estimated and true factors, and (2) Kernel Canonical Correlation Analysis (KCCA) for multiple latents, which evaluates the correlation between sets of estimated and true factors. An absolute value close to 1 indicates a strong correlation and better latent recovery. To evaluate the control performance, we measure the adaptation performance using: (3) jumpstart, which captures the improvement in initial performance when a learning agent leverages knowledge from source tasks; (4) accumulative reward, which indicates the learning quality over the

learning process; and (5) initial and final reward, which measures the initial performance benefiting from policy adaptation and the performance after the full training process.

**Baselines** For estimation evaluation, we compared with: (1) Disentangled Sequential Autoencoder [85], which disentangles latent representations into static and dynamic parts. However, it does not capture the global influence of $\kappa$, limiting its ability to model individual-specific factors. (2) Population-level component, which embeds latent factors based on population data instead of individual-specific information. For policy evaluation, our baselines include: (3) Aligned Latent Models [20], which jointly optimize a latent-space model and policy to maximize returns. (4) Soft Actor-Critic (SAC) [24], which incorporates entropy into the objective function to encourage exploration and improve robustness. (5) Deep Deterministic Policy Gradient (DDPG) [56], which combines deterministic policy gradients with deep neural networks for continuous action spaces. (6) Dueling Double Deep Q-Network (D3QN) [78], which introduces a dueling architecture for value function estimation to improve value estimation. (7) Rainbow DQN [28], which integrates prioritized experience replay and dueling network architectures to improve performance and learning stability.

### 6.1 Evaluation on Latent Estimation Framework

**Synthetic Experiments** We first conduct experiments on the synthetic dataset to evaluate the effectiveness of the estimation framework. The dataset is manually generated based on the post-nonlinear model. We design three types of latent factors $\kappa$, each either satisfying or violating the required assumptions. **Case 1**: $\kappa$ is a finite latent factor following the categorical distribution $\mathrm{Cat}(0.1, 0.2, 0.3, 0.4)$ with a cardinality of 4. **Case 2**: $\kappa$s are three-dimensional finite latent factors, each following a categorical distribution $\mathrm{Cat}(0.2, 0.8)$, $\mathrm{Cat}(0.2, 0.3, 0.5)$, $\mathrm{Cat}(0.1, 0.2, 0.3, 0.4)$, with cardinality equal to 2, 3, 4, respectively. **Case 3**: $\kappa$s are three-dimensional infinite latent factors, and each factor follows a Gaussian distribution $\mathcal{N}(0, 1)$, uniform distribution $\mathrm{Uniform}(0, 1)$, and exponential distribution $\mathrm{Exp}(1)$, respectively. We synthetically generate 40 unique trajectories, each representing an individual, with a maximum trajectory length of 20.

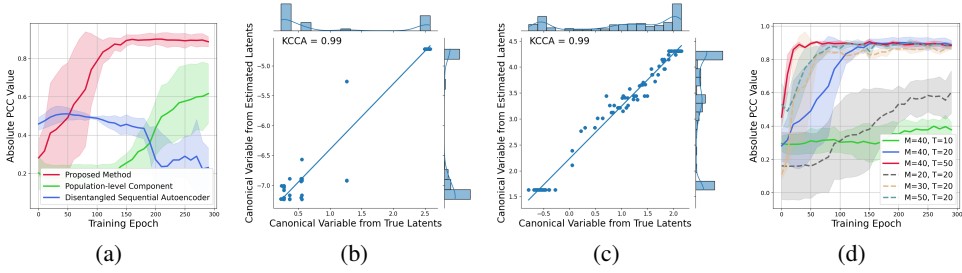

Figure 3: Synthetic results. (a) Comparisons of PCC trajectories in Case 1. (b-c) Scatterplot of the canonical variables in Cases 2 and 3. (d) Identifiability performance responses of the sample size.

For Case 1, we use PCC to evaluate estimation performance and report the training curve in Figure 3(a), where shaded regions indicate the standard deviation. The comparative results show that our method can recover the true latent factors, outperforming other baselines. Specifically, the population-level component fails to identify the individual-specific factor, as it overlooks the underlying differences between groups. Additionally, although the disentangled sequential autoencoder shows compromised identifiability in the early training stage by capturing the static part of the latent space, it fails to achieve full identifiability due to its inability to model individualized transition processes, leading to worse recovery performance over time. For Cases 2 and 3, we use KCCA to quality the correlation and visualize the results through scatter plots in Figure 3(b) and Figure 3(c). These results demonstrate identifiability in both finite and infinite latent factor scenarios and support the theoretical claims.

Moreover, we slightly violate the restriction on the number of individuals per group specified in Assumption 4.1 and analyze its impact on the training curve under varying population sizes, as shown by the dashed lines in Figure 3(d). The result shows that satisfying the sample sufficiency condition is necessary to recover the latent factors. In addition, we evaluate the effect of the trajectory length as outlined in Theorem 4.2. The findings, shown by the solid lines, demonstrate that increasing the

trajectory length would significantly improve the identifiability performance. This observation aligns with the theoretical guarantees provided by the proposed theorem.

**Ablation Study**   The contributions of the different components in the latent estimation framework are reported in Table 1. We build on the autoencoder framework with a quantization layer and add each component sequentially to the previous module. Incorporating a sequential encoder significantly improves the identifiability, which is important for the accurate recovery of latent factors. In the imple-

Table 1: Contribution of each module.

| Module | PCC $(\mu \pm \sigma)$ | Bias $(\mu \pm \sigma)$ |
|---|---|---|
| Quantized Encoding | 0.646 ± 3.1e-04 | 0.099 ± 2.3e-04 |
| + Sequential Encoder | 0.910 ± 1.3e-04 | 0.077 ± 5.7e-06 |
| + Noise Estimator | 0.942 ± 4.0e-05 | 0.072 ± 3.0e-07 |

mentation, we use a noise estimator during optimization to minimize bias and improve identifiability. The results suggest that the added components help fine-tuning the overall performance of the model, allowing for more accurate and reliable recovery of latent individual-specific factors.

**PersuasionForGood Corpus**   We further evaluate our framework on the real-world dataset, PersuasionForGood corpus [77], which is widely used for analyzing persuasion strategies [64, 7, 87]. The dataset contains 1017 person-to-person dialogues and 32 personality traits for each participant. In each dialogue, the persuader attempts to convince the persuadee to donate to a charity. In the context of iMDP, the state represents the persuadee's response, the action corresponds to the persuader's utterance, and the reward is defined as the final donation. Since this offline dataset does not include real-time interactions required for control performance evaluation, we focus on identifying the latent personality traits of each individual. We use BERT [9] as the backbone to embed each utterance into a 768-dimensional feature representation. These features are processed through an LSTM encoder, followed by a quantization layer, to recover the latent personality traits. The canonical correlation results under different latent dimensions are shown in Figure 4(a). The results demonstrate that our method is effective and feasible for real-world applications, particularly when the latent dimensions are fine-tuned appropriately.

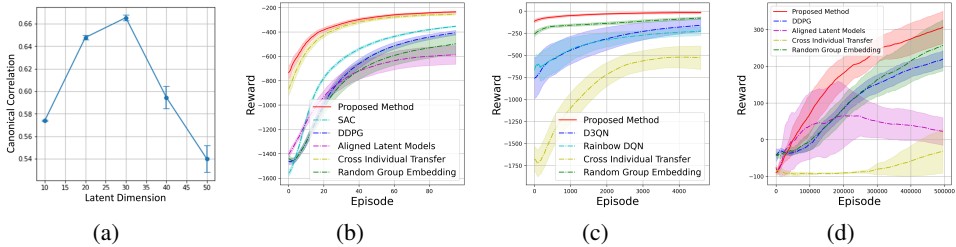

(a)                  (b)                  (c)                  (d)

Figure 4:   (a) Canonical correlation with respect to the latent dimensions in the PersuasionForGood corpus. (b-d) Accumulative reward curves in Pendulum, HeartPole, and Half Cheetah, respectively.

## 6.2   Evaluation on Policy Learning Framework

**Pendulum**   Pendulum [4] is a continuous control task for RL study with the goal of swinging up and stabilizing in an upright position. The states are the x-y coordinates and the angular velocity, and the action is the torque applied to the pendulum. For simplicity, we choose DDPG as the basic optimization algorithm and manually create 20 individualized environments. In these environments, the gravity $g$ is randomly drawn from a categorical distribution over the set $\{3, \ldots, 12\}$. The performance of the policy adaptation is evaluated on a new individual with $g = 10$.

We compare our method against several baselines: (1) SAC; (2) DDPG without prior knowledge; (3) aligned latent models; (4) pre-trained DDPG incorporating knowledge from given individuals, termed cross-individual transfer; (5) individualized policy incorporating randomly defined group embedding, termed random group embedding. The training curves over accumulative reward are reported in Figure 4(b), showing that the proposed method outperforms other baselines in both jumpstart and accumulative reward. Specifically, methods that benefit from population knowledge (our method and cross-individual transfer) outperform non-transfer methods, indicating that the pre-trained policy would accelerate the learning process. However, since cross-individual transfer ignores the individual-

specific information, such mixed policy knowledge yields worse initial performance compared to the individualized policies derived from our method.

**HeartPole**   HeartPole [57] is a discrete healthcare environment that explores the long-term health outcomes of short-term decisions. The six-dimensional states represent different health conditions, including alertness, hypertension, intoxication, time since sleep, time elapsed, and work done. Actions can be chosen from work, coffee, alcohol, and sleep. We create 100 individualized scenarios and assign each patient with individual characteristics, such as coffee tolerance, hypertension risk, and alcohol tolerance, according to a categorical distribution over the set $\{0.6, 0.8, 1.2\}$. The adaptation performance is evaluated on a new individual with all indices set to 1.

We compare our method against the following baselines: (1) D3QN without prior knowledge, (2) Rainbow DQN, (3) cross-individual transfer with D3QN, and (4) random group embedding. The training curves over accumulative reward are shown in Figure 4(c), and our method outperforms other baselines in both jumpstart and accumulative reward. Interestingly, while inappropriate source domain knowledge can negatively impact control performance (see cross-individual transfer), the result from random group embedding indicates that incorporating group embedding knowledge can enhance generalization. The group structure, together with properly estimated group information, jointly enables our method to converge better and faster than other baselines.

**Half Cheetah**   Half Cheetah [72] is a Mujoco-based task aiming to control a 2D bipedal robot. The agent consists of 9 links and 8 joints, and the goal is to apply torque to the joints to make the cheetah run forward as fast as possible. We introduced 50 individualized settings with the gravity $g$ following a categorical distribution with probabilities $p = 0.2$ and corresponding $g$ values from $\{8, 8.5, \ldots, 10\}$. The adaptation performance is evaluated on a new individual with $g = 9.8$. We compare our method with (1) DDPG without prior knowledge, (2) aligned latent models, (3) cross-individual transfer with DDPG, and (4) random group embedding. The comparative results, shown in Figure 4(d), indicate that our method outperforms all baselines in terms of convergence speed and efficiency. While inappropriate source domain data can degrade performance (see cross-individual transfer), the inclusion of group embedding facilitates generalization, enabling more effective adaptation.

## 7   Conclusion and Limitations

Our work focuses on learning latent state-transition processes from observed state-action trajectories, guaranteeing identifiability in the presence of latent individual-specific factors. To the best of our knowledge, this study provides novel identifiability guarantees for several settings that have not been previously addressed. Despite these contributions, our approach has three major limitations. (1) It currently does not account for instantaneous causal dependencies within $\mathbf{s}_t$. This limitation could be addressed by adjusting the temporal resolution of the data and explicitly modeling these dependencies. Integrating causal graphical models or advanced inference techniques could further enhance the framework's ability to handle instantaneous causal relationships. (2) While empirical results suggest the framework may adapt to scenarios with continuous latent factors, a formal nonparametric proof for these settings remains absent. Developing such a proof is an important avenue for future work. (3) The model assumes latent factors are time-invariant. Establishing such theoretical identifiability for time-varying latent factors is highly non-trivial and would require additional theoretical constraints to establish identifiability. These limitations highlight key directions for future research.

Furthermore, practical concerns such as privacy, robustness, and reliability are essential for real-world applications. To address privacy risks, techniques like de-identification (e.g., removing direct identifiers, data perturbation, pseudonymization) and differential privacy approaches should be explored. Ensuring privacy and security will enhance the framework's applicability in practice.

## Acknowledgement

This material is based upon work supported by NSF Award No. 2229881, AI Institute for Societal Decision Making (AI-SDM), the National Institutes of Health (NIH) under Contract R01HL159805, and grants from Salesforce, Apple Inc., Quris AI, and Florin Court Capital. BH is supported by NSF DMS-2428058.

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

# Supplementary Materials for "Identifying Latent State-Transition Processes for Individualized Reinforcement Learning"

## A  Notation and Terminology

We summarize the notations used throughout the paper in the following table.

| Index | |
|---|---|
| $\tau$ | Trajectory |
| $t$ | Time index |
| $T$ | Total length of time series |
| $G$ | Number of groups |
| $M$ | Number of individuals |
| $m$ | Index for a specific individual |
| $i, j$ | Variable element index |
| $\alpha, \beta$ | Weights of ELBO objective |
| $[G] = \{1, 2, \ldots, G\}$ | Sequence of integers from $1$ to $G$ inclusive |
| **Variable** | |
| $\epsilon_t$ | i.i.d. noise term for $\mathbf{s}$ at time $t$ |
| $f$ | State transition function |
| $\mathbf{s}_t, \hat{\mathbf{s}}_t$ | Observed & reconstructed states at time $t$ |
| $\mathbf{s}^m, \mathbf{a}^m, \kappa^m$ | State, action, and latent factor from individual $m$ |
| $\mathbf{s} = [s_1, s_2, \ldots, s_{d_s}]^\top$ | $d_s$-dimensional observed states |
| $\mathbf{a} = [a_1, a_2, \ldots, a_{d_a}]^\top$ | $d_a$-dimensional observed actions |
| $\kappa = [\kappa_1, \kappa_2, \ldots, \kappa_{d_\kappa}]^\top$ | $d_\kappa$-dimensional latent individual-specific factors |

## B  Identifiability Theory

Given the identifiability theorems, we first provide intuitive explanations for each assumption and discuss their relevance to real-world applications. Then, we provide the proof. Finally, we introduce some preliminaries related to our theorems, which are essential for the proof.

### B.1  Preliminaries for Theorem 4.1 and 4.2

#### B.1.1  Markov Property

The first-order Markov property implies that the transition probability to the next state depends only on the current state, uninfluenced by the sequence of previous states. Specifically,

**Definition B.1** (First-order Markov Property [70]). *A stochastic process $\{X_t : t \in \mathcal{N}\}$ has the first-order Markov property if, for each set of times $t, t-1, \ldots, 0$ and corresponding state $x_t, x_{t-1}, \ldots, x_0$ in the state space, the following conditional independence property holds:*

$$\mathbb{P}(X_t = x_t | X_{t-1} = x_{t-1}, X_{t-2} = x_{t-2}, \ldots, X_0 = x_0) = \mathbb{P}(X_t = x_t | X_{t-1} = x_{t-1}) \quad (5)$$

The first-order Markov property implies that the transition probability to the next state depends only on the current state, uninfluenced by the previous states. In the context of the state transition process, it possesses the first-order Markov property. Mathematically, it can be represented as:

$$\mathbb{P}(\mathbf{s}_t | \mathbf{s}_{t-1}, \mathbf{a}_{t-1}, \mathbf{s}_{t-2}, \mathbf{a}_{t-2}, \ldots, \mathbf{s}_0, \mathbf{a}_0) = \mathbb{P}(\mathbf{s}_t | \mathbf{s}_{t-1}, \mathbf{a}_{t-1}), \quad (6)$$

where $\mathbb{P}(\mathbf{s}_t | \mathbf{s}_{t-1}, \mathbf{a}_{t-1})$ is the transition probability from $(\mathbf{s}_{t-1}, \mathbf{a}_{t-1})$ to the state $\mathbf{s}_t$.

#### B.1.2  Finite Mixture Model

A finite mixture model is used for modeling a total population that comprises unobserved or hidden groups. Each of these groups is assumed to follow its own distinct probability distribution. In this context, the overall population model is expressed as a weighted sum of these individual distributions [61]. Specifically,

**Definition B.2** (Finite Mixture Models [74]). *A finite mixture model is a probability law based on a finite number of probability measures, $\mu_1, \ldots, \mu_m$, and a discrete distribution $\omega_1, \ldots, \omega_m$. A realization of a mixture model is generated by generating a component at random $k$, $1 \leq k \leq m$, and then drawing from $\mu_k \sim \mathcal{P}$. Then, the mixture measure $\mathcal{P}$ is defined as a weighted sum of probability measures $\mu_i$ with weights $w_i$. Specifically,*

$$\mathcal{P} = \sum_{i=1}^{m} w_i \delta_{\mu_i}. \tag{7}$$

## B.2 Discussions on Assumptions

**Group Determinacy**   The distinct values of the latent factor $\kappa$ categorize the population into separate groups, with each group characterized by its unique probability distribution and denoted as $\sum_{g=1}^{G} \pi_g \delta_{\kappa_g}(\kappa)$. The mixture formulation implies that the latent factor $\kappa$ serves as a categorical variable, with each unique value explicitly specifying a distinct group within the population. Such a formulation facilitates the identification and analysis of heterogeneous subpopulations within the finite mixture model.

The idea of group determinacy is important in real-world applications. Take personalized education as an example. For each student $m$, $\mathbf{s}^m$ represents their current knowledge state, $\mathbf{a}^m$ denotes their personalized learning action, and the function $f$ determines the unique educational trajectory for each student. The latent factor $\kappa^m$ influences how a student's learning progresses over time. It can be based on factors such as learning style preferences that help to logically group students. Specifically, one group might consist of visual learners who excel in interactive, graphically-oriented subjects, while another group might include students who prefer textual information and excel in reading and writing-intensive subjects. Each group exhibits its own set of learning outcomes and patterns, allowing educators to personalize teaching methods and materials to effectively meet the different needs of each group.

**Sample Sufficiency**   In a finite mixture model with $G$ groups, each group requires sufficient observations to identify the latent group factor $\kappa$. This assumption provides a minimum number of observation samples, which is $2G - 1$ observations in each group. Such a threshold ensures that we have enough information and variability in the observed data to distinguish the characteristics of each group. This assumption helps to identify the unique characteristics of each individual, which is critical for identifiability.

Sample sufficiency indicates that sufficient data are needed to achieve identifiability, which is a fundamental assumption in many analytical models. For example, in the context of nonlinear ICA using auxiliary variables [37], it is necessary to have at least $2n + 1$ values for the auxiliary variables to ensure sufficient variability and guarantee identifiability. Similarly, for successful disentanglement with minimal change [45], at least $2n + 1$ domain embeddings are required to ensure identifiability. Intuitively, without sufficient data to provide us with relevant information about the parameters, it is impossible to determine the values of these parameters.

**Asymptotic Identifiability**   Asymptotic identifiability refers to the property that a model becomes identifiable as the sample size goes to infinity. In practical terms, this means that given an infinite amount of data, one would be able to consistently estimate the parameters of the model. In the context of finite mixture models, if there are enough samples for each individual, then the corresponding components can be identified directly from each individual [35].

The overlapping condition requires the existence of at least two different individuals within the same group of the population, who have identical conditional probabilities. This assumption is crucial as it ensures that the model accounts for overlapping behavioral responses between different individuals, which is a common phenomenon in heterogeneous populations. Consider personalized education as an example, where students come from different academic backgrounds and have different levels of prior knowledge. Despite this initial heterogeneity, it is possible for two students in the same learning group to have the same probability of successfully completing a task.

## B.3 Proof of Theorem 4.1 and Theorem 4.2

We first show that the individualized transition processes can be viewed as a finite mixture model with grouped samples and then derive the identifiability under two scenarios.

*Proof on Theorem 4.1.* Assumptions in Theorem 4.1 and Eq. (2) ensure that individuals within the same group share identical joint distributions. Suppose the observations can be grouped into $G$ finite components, then the joint distribution can be factorized as:

$$\mathbb{P}(s, a, s') = \int \mathbb{P}(s, a, s'|u)\mathbb{P}(u)du \tag{8}$$

$$= \int \mathbb{P}(\kappa)\mathbb{P}(s', s, a|\kappa)d\kappa \tag{9}$$

This formulation asserts that the joint distribution of $\mathbb{P}(s, a, s')$ for the entire population can be modeled as a mixture model governed by the respective $\kappa_i$ values.

The following Lemma B.1 addresses the identifiability of mixture models from grouped samples.

**Lemma B.1** (Identifiability of Mixture Models from Grouped Samples [74])**.** *Suppose we have observations from a mixture model and that they are grouped such that observations in the same group are known to be drawn from the same component. Denote by $G$ the number of groups. If there are at least $2G - 1$ observations per group, any mixture of $G$ probability measures can be uniquely identified.*

It implies that with sufficient data per group, each component of the mixture model can be determined without ambiguity from the observed data. Specifically, suppose we have a mixture model consisting of $G$ different probability distributions, that is, $G$ components $c_1, c_2, \ldots, c_G$. Each component $c_i$ corresponds to a unique probability density function $\mathbb{P}_i(\cdot)$. These components are mixed together, with each component having a mixing weight $\pi_i$, satisfying $\pi_i \geq 0$ and $\sum_{i=1}^{G} \pi_i = 1$.

Now suppose we have $G$ observation groups $g_1, g_2, \ldots, g_G$, with each group $g_i$ containing observations that are independent and identically distributed drawn from the same component $c_i$. Since sample sufficiency ensures that the sample size of observations within each group is greater than $2G - 1$, then the identifiability of $\kappa$ is guaranteed by Lemma B.1. □

*Proof on Theorem 4.2.* Under the assumptions in Theorem 4.2, if each individual's trajectory is sufficiently long, the latent individual-specific factors become asymptotically identifiable. Consider an extreme scenario where each individual is treated as a distinct group. Some individuals may share the same latent factor $\kappa$ and can be grouped together. In that case, it is necessary to measure the similarity between individuals and merge those with the same $\kappa$ into a single group.

An intuitive criterion for merging is as follows. Asymptotically, for a particular state-action pair $(\mathbf{s}, \mathbf{a}) = (\mathbf{s}^*, \mathbf{a}^*)$, the probability of $\mathbf{s}'$ given $(\mathbf{s}^*, \mathbf{a}^*)$ will be identical across individuals in the same group. Define $t^j$ as the time of the $j$-th occurrence of $(\mathbf{s}^*, \mathbf{a}^*)$. For each $j$, let $\mathbf{X}^j = \{\mathbf{s}_{t^j+1}, t^j = 1, \ldots\}$, representing the collection of states observed at time $t + 1$ given the fixed state-action pair $(\mathbf{s}^*, \mathbf{a}^*)$ at time $t$. In this way, $\mathbf{X}^j$ can be considered as samples drawn from a particular group $j$. Identifiability is then guaranteed by Lemma B.1, since the distributions of $\mathbf{X}^j$ allow the grouping of individuals based on shared latent factors.

□

**Remark B.1.** *Prior work [35] used a Gaussian mixture model as a prior on the coefficients, while the latent confounder variable, denoted as $Z$, was constrained to a binary state, thereby indicating group membership for a given individual. We extend this foundation by generalizing the latent confounder to a set of discrete values and considering a nonparametric model for broader applications.*

## C  Further Discussion on Identifiability Theorem

Recent work [10, 34] provides the necessary and sufficient conditions for the identifiability of certain latent structural patterns, but it rules out the case of triangle structure involving latent variables. In

this paper, we extend their work to the temporal case and provide the identifiability of the latent group factor $\kappa$, where $\kappa$ can be either continuous or discrete. Furthermore, we allow multiple instances of $\kappa$ to influence the state transition dynamics.

## C.1 Problem Setting

In Theorem 4.3, we aim to identify the latent group factors based on a post-nonlinear temporal causal model, as shown in Figure 5. We assume the existence of a learnable and invertible embedding mapping $f$, which is able to preserve the causal structure intrinsic to the state $s$. Specifically,

**Definition C.1** (Post-nonlinear Temporal Causal Models). *Consider a scenario where $d_s$ observed states from the $k$-th individual are denoted as $\mathbf{s}_t^k = (s_{1,t}^k, \ldots, s_{d_s,t}^k)^{\mathrm{T}}$, which is a direct observation of an embedded representation $h(s)$, alongside $m$ unobserved group factors $\kappa = (\kappa_1, \ldots, \kappa_m)^{\mathrm{T}}$. The state transition dynamics satisfy*

$$s_{i,t+1}^k = h^{-1}\left(\sum_{j \in \mathcal{P}_i} \alpha_{ij} h(s_{j,t}^k) + \sum_{j \in \mathcal{L}_i} \beta_{ij} a_{j,t}^k + \sum_{j=1}^m \lambda_j \kappa_j + \epsilon_{i,t+1}^k\right), \tag{10}$$

*for $i = 1, \ldots, n$. Here, $\alpha_{ij}$ and $\beta_{ij}$ represent causal coefficients that quantify the influence of the state $h(s_{j,t})$ and the action $a_{j,t}$ on $h(s_{i,t+1})$, respectively. $\mathcal{P}_i$ and $\mathcal{L}_i$ denote the sets of direct state and action that influences $s_{i,t+1}^k$ (or $h(s_{i,t+1}^k)$). Actions are considered to be stochastic. The coefficients $\lambda_j$ are individual-specific and show variation across individuals. The random noise term $\epsilon_{i,t+1}^k$ is independent of $s_{j,t}$ and $a_{j,t}$ for all $j \in \mathcal{N}^+$ to account for unmeasured influences.*

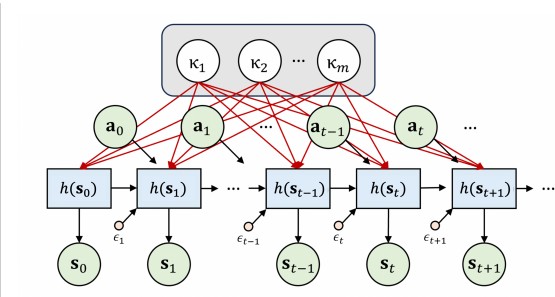

Figure 5: Post-nonlinear temporal causal model.

**Definition C.2** (Minimal Rank Set). *Let $\mathcal{S}_t = \{s_{1,t}, s_{2,t}, \ldots, s_{d_s,t}\}$ represent the set of all state variables in the system at any time $t = 1, \ldots, T$, and let $\mathcal{L} = \{\kappa_1, \kappa_2, \ldots, \kappa_m\}$ represent the set of latent variables, where $m$ and $d_s$ are the numbers of latent and state dimensions, respectively. A subset $\mathcal{R}_{t,t^-} \subseteq \mathcal{S}_t \cup \mathcal{S}_{<t}$ (or $\mathcal{R}_{t,t^+} \subseteq \mathcal{S}_t \cup \mathcal{S}_{>t}$) with cardinality $r$, is called a minimal rank set if it satisfies the following conditions:*

(i) *The bottleneck set, defined as $\mathcal{B} = \mathcal{L} \cup \mathcal{S}_t$ for any given time $t$, can t-separate (see Definition C.5) any pair of minimal rank sets $(\mathcal{R}_{t,t^-}, \mathcal{R}_{t,t^+})$, where $t^- < t < t^+$.*

(ii) *There does not exist a subset $\mathcal{R}'_{t,t^\pm} \subset \mathcal{R}_{t,t^\pm}$ with $|\mathcal{R}'_{t,t^\pm}| < |\mathcal{R}_{t,t^\pm}|$ that can satisfy condition (i).*

The set $\mathcal{R}_{t,t^\pm}$ is considered *minimal* in the sense that it is the smallest cardinality subset of observed state variables that includes a bottleneck set and disjoint state variables, capable of representing the essential separation status within the system. An illustrative example of a minimal rank set (see Figure 6) is shown in the yellow area, and a bottleneck set is depicted in the green area.

## C.2 Proof of Theorem 4.3

The underlying intuition of Theorem 4.3 is that, in the absence of latent variables, rank information should align with what Conditional Independence (CI) skeleton (see Definition C.6) provides; if not, then there must exist at least one latent variable.

### C.2.1 Necessary Lemmas

The following lemma indicates that the rank of the covariance matrix (see Definition C.3) $\Sigma_{\mathbf{A},\mathbf{B}}$ between any two sets of variables $\mathbf{A}$ and $\mathbf{B}$ is less than or equal to the sum of cardinalities of any trek-separating (see Definition C.5) sets $\mathbf{C_A}$ and $\mathbf{C_B}$. The equality holds for generic covariance matrices consistent with the graph $\mathcal{G}$.

**Lemma C.1** (Trek Separation for Directed Graphical Models [68]). *The submatrix $\Sigma_{\mathbf{A},\mathbf{B}}$ has rank less than or equal to $r$ for all covariance matrices consistent with the graph $\mathcal{G}$ if and only if there exist subsets $\mathbf{C_A}, \mathbf{C_B} \subset V(\mathcal{G})$ with $|\mathbf{C_A}| + |\mathbf{C_B}| \leq r$ such that $\mathbf{C_A}, \mathbf{C_B}$ t-separates $\mathbf{A}$ from $\mathbf{B}$. Consequently,*

$$\text{rank}(\Sigma_{\mathbf{A},\mathbf{B}}) \leq \min\{|\mathbf{C_A}| + |\mathbf{C_B}| : (\mathbf{C_A}, \mathbf{C_B}) \text{ t-separates } \mathbf{A} \text{ from } \mathbf{B}\} \tag{11}$$

*and equality holds for generic covariance matrices consistent with $\mathcal{G}$.*

**Lemma C.2** (Identifiability of Linear Regression Models). *Consider a linear regression model with a response variable $Y$ and $p$ predictors $X_1, X_2, \ldots, X_p$. The linear relationship is defined as:*

$$Y = \beta_0 + \beta_1 X_1 + \beta_2 X_2 + \ldots + \beta_p X_p + \varepsilon \tag{12}$$

*where $\beta_0, \beta_1, \ldots, \beta_p$ are the regression coefficients and $\varepsilon$ is the error term. The matrix representation can be expressed as:*

$$\mathbf{Y} = \mathbf{X}\boldsymbol{\beta} + \boldsymbol{\varepsilon} \tag{13}$$

*where $\mathbf{Y}$ is the response vector, $\mathbf{X}$ is the design matrix including predictors, $\boldsymbol{\beta}$ is the vector of regression coefficients, and $\boldsymbol{\varepsilon}$ is the vector of error terms. For the regression coefficients $\boldsymbol{\beta}$ to be identifiable, the design matrix $\mathbf{X}$ must have full column rank, meaning no predictor is a perfect linear combination of the others. This ensures that the matrix $\mathbf{X}^T \mathbf{X}$ is invertible, allowing for the unique estimation of $\boldsymbol{\beta}$ through:*

$$\hat{\boldsymbol{\beta}} = (\mathbf{X}^T \mathbf{X})^{-1} \mathbf{X}^T \mathbf{Y} \tag{14}$$

**Lemma C.3** (Identifiability of Factor Analysis). *Consider a factor analysis model with $p$ observations for each of $n$ individuals and $k$ common factors ($k < p$). The relationship is defined by the factor loading matrix $L \in \mathbb{R}^{p \times k}$ and the factor matrix $F \in \mathbb{R}^{k \times n}$. Specifically,*

$$X = LF + \varepsilon \tag{15}$$

*where $X \in \mathbb{R}^{p \times n}$ is the observation matrix and $\varepsilon \in \mathbb{R}^{p \times n}$ is the error term matrix. The factor loading matrix $L$ and the factor matrix $F$ are unique up to an orthogonal transformation. Specifically, for any orthogonal matrix $Q$, if we set $L' = LQ$ and $F' = Q^T F$, the transformed matrices $L'$ and $F'$ also satisfy the model criteria.*

### C.2.2 Proof of Structure Identifiability

*Proof.* Suppose latent factors exist and influence the embedding of the observed states $h(s)$, which preserve the causal structure intrinsic to the states $s$. According to Lemma C.1, the rank of $\Sigma_{\mathbf{A_i},\mathbf{B_i}}$ should be less than or equal to $\min\{|\mathbf{C_{A_i}}| + |\mathbf{C_{B_i}}|\}$. In the absence of latent factors, according to the CI skeleton, the minimal configuration to t-separate $\mathbf{A_i}$ from $\mathbf{B_i}$ is by

$$(\{h(s_{1,i}), \ldots, h(s_{d_s,i})\}, \emptyset) \quad \text{or} \quad (\emptyset, \{h(s_{1,i}), \ldots, h(s_{d_s,i})\}). \tag{16}$$

Consequently, the rank of the covariance matrix is $\text{rank}(\Sigma_{\mathbf{A_i},\mathbf{B_i}}) = |\{h(s_{1,i}), \ldots, h(s_{d_s,i})\}| + |\emptyset| = d_s$. If the calculated rank is greater than $d_s$, it implies the presence of latent variables accounting for the unexplained variance since the observed variables alone would not result in such rank deficiency.

In scenarios with latent factors, the maximum rank deficiency observed across covariance submatrices, representing the discrepancy between the expected and actual ranks, establishes a lower bound for the number of latent variables. Considering the minimal t-separation of $\mathbf{A_i}$ and $\mathbf{B_i}$ occurs via

$$(\{h(s_{1,i}), \ldots, h(s_{d_s,i}), \kappa_1, \ldots, \kappa_m\}, \emptyset) \quad \text{or} \quad (\emptyset, \{h(s_{1,i}), \ldots, h(s_{d_s,i}), \kappa_1, \ldots, \kappa_m\}). \tag{17}$$

Then the rank of the covariance matrix is $\text{rank}(\Sigma_{\mathbf{A_i},\mathbf{B_i}}) = |\{h(s_{1,i}), \ldots, h(s_{d_s,i}), \kappa_1, \ldots, \kappa_m\}| + |\emptyset| = m + d_s$. By iteratively computing the rank of $\text{rank}(\Sigma_{\mathbf{A_i},\mathbf{B_i}})$, a consistent value corroborates the existence of latent factors influencing all observed states. Furthermore, the count of latent factors can be deduced by $m = \text{rank}(\Sigma_{\mathbf{A_i},\mathbf{B_i}}) - d_s$.

In conclusion, under the conditions of the theorem, if the observed rank deficiency in the covariance matrix of observed variables cannot be explained by the observed variables alone, it implies the existence of latent variables. Furthermore, the number of such latent variables can be inferred from the extent of the rank deficiency. $\square$

### C.2.3 Proof of Parameter Identifiability

*Proof.* For each individual $k$, consider the proposed model at any time $t$ and $t+1$:

$$h(s_{i,t}^k) = \sum_{j \in \mathcal{P}_i} \alpha_{ij} h(s_{j,t-1}^k) + \sum_{j \in \mathcal{L}_i} \beta_{ij} a_{j,t-1}^k + \sum_{j=1}^m \lambda_j \kappa_j + \epsilon_{i,t}^k,$$

$$h(s_{i,t+1}^k) = \sum_{j \in \mathcal{P}_i} \alpha_{ij} h(s_{j,t}^k) + \sum_{j \in \mathcal{L}_i} \beta_{ij} a_{j,t}^k + \sum_{j=1}^m \lambda_j \kappa_j + \epsilon_{i,t+1}^k.$$

Subtracting these two equations, we obtain:

$$h(s_{i,t+1}^k) - h(s_{i,t}^k) = \sum_{j \in \mathcal{P}_i} \alpha_{ij}(h(s_{j,t}^k) - h(s_{j,t-1}^k)) + \sum_{j \in \mathcal{L}_i} \beta_{ij}(a_{j,t}^k - a_{j,t-1}^k) + (\epsilon_{i,t+1}^k - \epsilon_{i,t}^k).$$

Define $x_{i,t+1}^k = h(s_{i,t+1}^k) - h(s_{i,t}^k)$, $y_{i,t+1}^k = a_{i,t+1}^k - a_{i,t}^k$, and $\eta_{i,t+1}^k = \epsilon_{i,t+1}^k - \epsilon_{i,t}^k$. Substituting these, the model transforms to:

$$x_{i,t+1}^k = \sum_{j \in \mathcal{P}_i} \alpha_{ij} x_{j,t}^k + \sum_{j \in \mathcal{L}_i} \beta_{ij} y_{j,t}^k + \eta_{i,t+1}^k.$$

In that case, the identifiability of $\alpha$ and $\beta$ can be directly derived by Lemma C.2. We further assume that the $m$ latent factors follow the Normal distribution. Drawing on methodologies used in factor analysis C.3, then $\lambda$ is orthogonal-wise identifiable. $\square$

### C.3 Examples For Theorem 4.3

In this section we present four illustrative examples to describe cases where identifiability is achieved. For the sake of simplicity, we define $X$ as $X = h(s)$ and omit the terms $a$ and $\epsilon$ from our illustration for simplicity, under the assumption that they are random and independent variables.

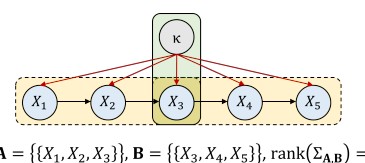

$\mathbf{A} = \{\{X_1, X_2, X_3\}\}, \mathbf{B} = \{\{X_3, X_4, X_5\}\}, \text{rank}(\Sigma_{\mathbf{A,B}}) = 2$

Figure 6: Single latent variable and one-dimension state.

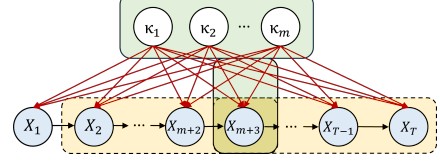

Figure 7: Multiple latent variables and one-dimension state.

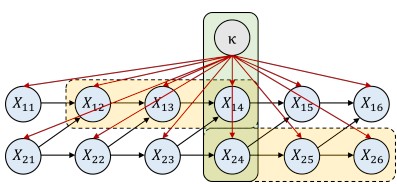

Figure 8: Single latent variable and multi-dimension states.

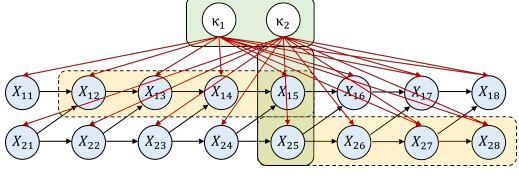

Figure 9: Multiple latent variables and multi-dimension states.

Figure 10: Examples that illustrate different identifiable cases.

**Example 1** In Example 1, as shown in Figure 6, there is only one latent factor and one-dimensional states. The bottleneck set is $(X_3, \kappa)$, and the pairs of minimal rank sets are $(\mathbf{A}, \mathbf{B}) = ((X_1, X_2, X_3), (X_3, X_4, X_5))$. According to the causal graph, the minimal way to t-separate $\mathbf{A}$ from $\mathbf{B}$ is either $(\{\kappa, X_3\}, \emptyset)$ or $(\emptyset, \{\kappa, X_3\})$. Consequently, the rank of the covariance matrix is $\text{rank}(\Sigma_{\mathbf{A,B}}) = |\{\kappa, X_3\}| + |\emptyset| = 2$. According to Theorem 2, the fact that $\text{rank}(\Sigma_{\mathbf{A,B}}) = 2 > 1$ indicates the presence of the latent factor. Consequently, the number of latent variables can be deduced as $m = \text{rank}(\Sigma_{\mathbf{A,B}}) - 1 = 2 - 1 = 1$.

**Example 2** As shown in Figure 7, there are $m$ latent factors and one-dimensional states. The bottleneck sets are $((X_{m+2}, \kappa_1, \ldots, \kappa_m), \ldots, (X_{T-m-1}, \kappa_1, \ldots, \kappa_m))$. Suppose $T = 2m + 4$, then the pairs of minimal rank sets are $(\mathbf{A_1}, \mathbf{B_1}) = ((X_1, \ldots, X_{m+2}), (X_{m+2}, \ldots, X_{2m+3}))$ and $(\mathbf{A_2}, \mathbf{B_2}) = ((X_2, \ldots, X_{m+3}), (X_{m+3}, \ldots, X_{2m+4}))$. According to the graph, the minimal configuration to t-separate $\mathbf{A_1}$ from $\mathbf{B_1}$ is either $(\{\kappa_1, \ldots, \kappa_m, X_{m+2}\}, \emptyset)$ or $(\emptyset, \{\kappa_1, \ldots, \kappa_m, X_{m+2}\})$ with $\mathrm{rank}(\Sigma_{\mathbf{A_1},\mathbf{B_1}}) = |\{\kappa_1, \ldots, \kappa_m, X_{m+2}\}| + |\emptyset| = m + 1$. The minimal configuration to t-separate $\mathbf{A_2}$ from $\mathbf{B_2}$ is either $(\{\kappa_1, \ldots, \kappa_m, X_{m+3}\}, \emptyset)$ or $(\emptyset, \{\kappa_1, \ldots, \kappa_m, X_{m+3}\})$, resulting in $\mathrm{rank}(\Sigma_{\mathbf{A_2},\mathbf{B_2}}) = |\{\kappa_1, \ldots, \kappa_m, X_{m+3}\}| + |\emptyset| = m + 1$. According to Theorem 2, the fact that $\mathrm{rank}(\Sigma_{\mathbf{A_1},\mathbf{B_1}}) = \mathrm{rank}(\Sigma_{\mathbf{A_2},\mathbf{B_2}}) = m + 1 > 1$ indicates the presence of the latent factor. Consequently, the number of latent variables can be deduced as $m = \mathrm{rank}(\Sigma_{\mathbf{A_i},\mathbf{B_i}}) - 1 = m + 1 - 1 = m$.

**Example 3** As shown in Figure 8, there is one latent factor and two-dimensional states. The bottleneck sets are $((X_{13}, X_{23}, \kappa), (X_{14}, X_{24}, \kappa))$, and one possible pairs of minimal rank sets are $(\mathbf{A_1}, \mathbf{B_1}) = ((X_{11}, X_{12}, X_{13}, X_{23}), (X_{13}, X_{23}, X_{24}, X_{25}))$ and $(\mathbf{A_2}, \mathbf{B_2}) = ((X_{12}, X_{13}, X_{14}, X_{24}), (X_{14}, X_{24}, X_{25}, X_{26}))$. According to the causal graph, the minimal configuration to t-separate $\mathbf{A_1}$ from $\mathbf{B_1}$ is either $(\{\kappa, X_{13}, X_{23}\}, \emptyset)$ or $(\emptyset, \{\kappa, X_{13}, X_{23}\})$. Consequently, the rank of the covariance matrix is $\mathrm{rank}(\Sigma_{\mathbf{A_1},\mathbf{B_1}}) = |\{\kappa, X_{13}, X_{23}\}| + |\emptyset| = 3$. Similarly, the minimal configuration to t-separate $\mathbf{A_2}$ from $\mathbf{B_2}$ is either $(\{\kappa, X_{14}, X_{24}\}, \emptyset)$ or $(\emptyset, \{\kappa, X_{14}, X_{24}\})$, resulting in $\mathrm{rank}(\Sigma_{\mathbf{A_2},\mathbf{B_2}}) = |\{\kappa, X_{14}, X_{24}\}| + |\emptyset| = 3$. According to Theorem 2, the fact that $\mathrm{rank}(\Sigma_{\mathbf{A_1},\mathbf{B_1}}) = \mathrm{rank}(\Sigma_{\mathbf{A_2},\mathbf{B_2}}) = 3 > 2$ indicates the presence of the latent factor. Consequently, the number of latent variables can be deduced as $m = \mathrm{rank}(\Sigma_{\mathbf{A_i},\mathbf{B_i}}) - 2 = 3 - 2 = 1$.

**Example 4** As shown in Figure 9, there are two latent factors and two-dimensional states. The bottleneck sets are $((X_{14}, X_{24}, \kappa_1, \kappa_2), (X_{15}, X_{25}, \kappa_1, \kappa_2))$, and one possible pairs of minimal rank sets are $(\mathbf{A_1}, \mathbf{B_1}) = ((X_{11}, X_{12}, X_{13}, X_{14}, X_{24}), (X_{14}, X_{24}, X_{25}, X_{26}, X_{27}))$ and $(\mathbf{A_2}, \mathbf{B_2}) = ((X_{12}, X_{13}, X_{14}, X_{15}, X_{25}), (X_{15}, X_{25}, X_{26}, X_{27}, X_{28}))$. According to the graph, the minimal configuration to t-separate $\mathbf{A_1}$ from $\mathbf{B_1}$ is either $(\{\kappa_1, \kappa_2, X_{14}, X_{24}\}, \emptyset)$ or $(\emptyset, \{\kappa_1, \kappa_2, X_{14}, X_{24}\})$. Consequently, the rank of the covariance matrix is $\mathrm{rank}(\Sigma_{\mathbf{A_1},\mathbf{B_1}}) = |\{\kappa_1, \kappa_2, X_{14}, X_{24}\}| + |\emptyset| = 4$. Similarly, the minimal configuration to t-separate $\mathbf{A_2}$ from $\mathbf{B_2}$ is either $(\{\kappa_1, \kappa_2, X_{15}, X_{25}\}, \emptyset)$ or $(\emptyset, \{\kappa_1, \kappa_2, X_{15}, X_{25}\})$ with $\mathrm{rank}(\Sigma_{\mathbf{A_2},\mathbf{B_2}}) = |\{\kappa_1, \kappa_2, X_{15}, X_{25}\}| + |\emptyset| = 4$. According to Theorem 2, the fact that $\mathrm{rank}(\Sigma_{\mathbf{A_1},\mathbf{B_1}}) = \mathrm{rank}(\Sigma_{\mathbf{A_2},\mathbf{B_2}}) = 4 > 2$ indicates the presence of the latent factor. Consequently, the number of latent variables can be deduced as $m = \mathrm{rank}(\Sigma_{\mathbf{A_i},\mathbf{B_i}}) - 2 = 4 - 2 = 2$.

## C.4 Related Definitions of Theorem 4.3

### C.4.1 Covariance Matrix of Random Vector

In this discussion, we introduce the concept of the covariance matrix within the framework of latent variable models. By examining the properties of the covariance matrix, such as its rank, we are able to identify signs of latent variables—rank deficiencies, which serve as a measure of the cardinality of the minimal set of latent variables required to explain the observed dependencies. Such rank deficiencies indicate the presence of latent variables that extend beyond the observable scope.

Consider a directed acyclic graph (DAG), denoted as $\mathcal{G}$, whose vertices $V(\mathcal{G})$ form the set $[m] := \{1, 2, \ldots, m\}$. Each node $i$ in $\mathcal{G}$ is associated with a random variable $X_i$ and an independent error term $\epsilon_i \sim \mathcal{N}(0, \phi_i)$ with $\phi_i > 0$. The DAG structure imposes a recursive relationship among the variables, where the value of $X_j$ can be expressed as a linear combination of the variables $X_i$ of its parent vertices $\mathrm{pa}(j)$, alongside the error term $\epsilon_j$ and regression coefficients $\lambda_{ij}$ that correspond to the edges $i \to j$ in $\mathcal{G}$:

$$X_j = \sum_{i \in \mathrm{pa}(j)} \lambda_{ij} X_i + \epsilon_j. \tag{18}$$

where $\mathrm{pa}(j)$ denotes the set of parent nodes of vertex $j$, where a parent node $i$ is one that has an edge leading to $j$ in $\mathcal{G}$. From this recursive sequence of regressions, one can solve for the covariance matrix $\Sigma$ of the jointly normal random vector $\mathbf{X}$, which is defined as follows.

**Definition C.3** (Covariance Matrix of Random Vector [68])**.** *The covariance matrix of the random vector is given by the matrix factorization*

$$\Sigma = \Lambda^{-\top} \Phi \Lambda^{-1}. \tag{19}$$

*where matrix $\Phi$ is defined as a diagonal matrix with the variances of the error terms as its diagonal elements:* $\Phi = \mathrm{diag}(\phi_1, \ldots, \phi_m)$. *The matrix $M$ is an $m \times m$ upper triangular matrix where* $M_{ij} = \lambda_{ij}$ *if $i \to j$ is an edge in $\mathcal{G}$, and $M_{ij} = 0$ otherwise. Thus the matrix $\Lambda$ is defined as* $\Lambda = I - M$, *where $I$ is the $m \times m$ identity matrix.*

Specifically, given two subsets $\mathbf{A}, \mathbf{B} \subset [m]$, $\Sigma_{\mathbf{A},\mathbf{B}} = (\sigma_{ab})_{a \in \mathbf{A}, b \in \mathbf{B}}$ is defined as the submatrix of covariance with row index set $\mathbf{A}$ and column index set $\mathbf{B}$.

### C.4.2 Trek and Trek Separation

The concepts of Trek and Trek Separation precede a crucial need to address the presence of latent variables and intricate dependency structures that are not directly observable. The Trek represents a particular path that interconnects variables within a graph, even if they are not directly linked, while Trek Separation delineates a criterion to ascertain whether two sets of variables are independent, conditional on a set of other variables. Below, we give the formation definitions of these two concepts.

**Definition C.4** (Trek [68]). *A trek in $\mathcal{G}$ from $i$ to $j$ is an ordered pair of directed paths $(P_1, P_2)$ where $P_1$ has sink $i$, $P_2$ has sink $j$, and both $P_1$ and $P_2$ have the same source $k$. The common source $k$ is called the top of the trek, denoted $\mathrm{top}(P_1, P_2)$. Note that one or both of $P_1$ and $P_2$ may consist of a single vertex, that is, a path with no edges. A trek $(P_1, P_2)$ is simple if the only common vertex among $P_1$ and $P_2$ is the common source $\mathrm{top}(P_1, P_2)$. We let $\mathcal{T}(i, j)$ and $\mathcal{S}(i, j)$ denote the sets of all treks and all simple treks from $i$ to $j$, respectively.*

**Definition C.5** (Trek Separation [68]). *Let $\mathbf{A}, \mathbf{B}, \mathbf{C_A}$ and $\mathbf{C_B}$ be four subsets of $V(\mathcal{G})$ which need not be disjoint. We say that the pair $(\mathbf{C_A}, \mathbf{C_B})$ trek separates (or $t$-separates) $\mathbf{A}$ from $\mathbf{B}$ if for every trek $(P_1, P_2)$ from a vertex in $\mathbf{A}$ to a vertex in $\mathbf{B}$, either $P_1$ contains a vertex in $\mathbf{C_A}$ or $P_2$ contains a vertex in $\mathbf{C_B}$.*

### C.4.3 Conditional Independence Skeleton

The Conditional Independence (CI) skeleton in graphical models refers to a structure that represents the conditional independence among observed variables. The CI skeleton can be used to infer the existence of latent variables. If the observed data suggests dependencies not represented in the CI skeleton, it may indicate hidden factors at play. The formal definition is given as follows.

**Definition C.6** (Conditional Independence Skeleton [10]). *A CI skeleton of $\mathbf{X}$ is an undirected graph where the edge between $X_1$ and $X_2$ exists if and only if there does not exist a set of observed variables $\mathbf{C}$ such that $X_1, X_2 \notin \mathbf{C}$ and $X_1 \perp\!\!\!\perp X_2 | \mathbf{C}$.*

## D  Background

**Reinforcement Learning**    In RL, an agent learns to make decisions by interacting with the environment. The agent receives rewards for taking actions in the environment and uses this feedback to learn optimal behavior. It is often modeled as a Markov Decision Process (MDP) represented by a tuple $\langle \mathcal{S}, \mathcal{A}, \mathbb{P}, R, \gamma \rangle$, where $\mathcal{S}$ denotes a finite set of states representing different situations an agent might encounter, $\mathcal{A}$ a finite set of actions representing different decisions an agent can make, $\mathbb{P}$ a state transition function defining the probability of transitioning to a new state $s'$ given a current state $s$ and action $a$, denoted as $\mathbb{P}(s'|s,a)$, $R$ a reward function assigning a scalar value to each state-action pair $(s, a)$, representing the immediate reward received after performing action $a$ in state $s$. $\gamma \in [0, 1]$ is the discount factor, representing the agent's consideration for future rewards. The agent's goal is to learn an optimal policy $\pi^*$, which defines the optimal set of actions in different states to maximize the expected cumulative discounted reward over the long run. Developing this optimal policy involves estimating value functions such as the action-value function, defined as $Q^\pi(s, a) = \mathbb{E}_\pi \left[ \sum_{t=0}^{\infty} \gamma^t R_t | S_0 = s, A_0 = a \right]$, which represents the expected reward of taking action $a$ in state $s$ following policy $\pi$. The pursuit of optimal policy $\pi^*$ involves maximizing the value functions over all possible state-action pairs: $\pi^* = \arg\max_\pi Q^\pi(s, a)$.

**Variational Autoencoder**    Variational Autoencoders (VAEs) [42] are a class of generative models in deep learning, adept at unsupervised learning of complex data distributions. Rooted in the framework of Bayesian inference, VAEs are designed to approximate probability density functions of input data. The architecture of a VAE consists of two primary components: an encoder $q_\phi(z|x)$ and a decoder

$p_\theta(x|z)$. The encoder maps input data $x$ to a latent space, represented by a probability distribution, typically Gaussian, with parameters $\mu$ and $\sigma$ signifying the mean and standard deviation, respectively. The decoder reconstructs the input data from a sampled latent representation $z$.

The distinct feature of VAEs lies in their probabilistic approach. The encoder outputs parameters of a latent distribution, from which a sample $z$ is drawn:

$$z \sim q_\phi(z|x) = \mathcal{N}(z; \mu, \sigma^2 I) \tag{20}$$

The decoder then attempts to reconstruct the input from this latent sample. VAEs optimize the Evidence Lower Bound (ELBO) objective, which balances two aspects: the reconstruction quality and the regularization of the latent space. The traditional ELBO is given by:

$$\text{ELBO} = \mathbb{E}_{q_\phi(z|x)}[\log p_\theta(x|z)] - \text{KL}[q_\phi(z|x)||p(z)] \tag{21}$$

Here, the first term measures the reconstruction quality, while the second term, the Kullback-Leibler (KL) divergence, imposes a regularization by encouraging the latent distribution $q_\phi(z|x)$ to be close to a prior $p(z)$, typically assumed to be a standard normal distribution $\mathcal{N}(0, I)$. VAEs, through this optimization, are capable of generating new data points that are similar to the input data, making them highly valuable in applications like image generation, denoising, and anomaly detection within the domain of unsupervised learning.

# E    Detailed Related Work

**Individualized Machine-Learning Applications**    Machine learning has been increasingly leveraged to create highly individualized solutions in a variety of domains. In health and wellness, it supports tailored interventions to increase physical activity [86, 55], promote weight loss [16, 15], and improve adherence to diabetes management [86]. For the elderly, personalized algorithms facilitate technology adaptation and specialized care for specific conditions [30]. In the financial sector, machine learning improves the optimization of technical indicators and stock market predictions, making them more individualized [54]. In education, Information and Communication Technology (ICT) uses machine learning to provide personalized education systems such as adaptive e-learning systems [14] and individualized tutorial planning [38]. The transportation sector benefits from individualized car-following control strategies designed for specific driver behaviors [67]. Furthermore, multimedia platforms, such as YouTube and TikTok, are enhancing the user experience by using reinforcement learning to recommend video content tailored to individual preferences [5, 31]. These examples underscore the breadth of individualized machine-learning applications, highlighting their transformative impact across various industries.

**Reinforcement Learning for Latent State-Transition Processes**    RL has witnessed significant advancements in recent years, particularly through the integration of latent variable models to capture the underlying dynamics of environments. A primary focus in this domain is learning low-dimensional, latent Markovian representations from observed data [52, 46, 40, 23, 79, 92, 47, 60, 17, 21, 18, 88]. Common strategies for state representation learning include reconstructing the observations, learning forward models, or learning inverse models. Additionally, prior knowledge, such as temporal continuity [80], is often used to constrain the state space. Numerous studies have proposed methods to estimate the underlying state-transition processes from high-dimensional input sequences [79, 11, 23, 25, 91, 17, 39, 26]. Leveraging these learned world models, agents can perform model-based RL or planning. Such methods typically encode structural constraints, ensuring the sufficiency and minimality of the estimated state representations from both generative and selection processes. Recently, there has been growing interest in estimating state-transition processes in the presence of latent confounders [58, 53, 75, 76, 3, 63]. Some studies [58, 63] address similar settings involving individual-specific factors. However, to the best of our knowledge, we have yet to identify a systemic approach that provides clear identifiability guarantees for the state-transition processes in the presence of individual-specific latent factors.

**Comparisons with Related Works**    Existing methods for modeling latent factors in Markov Decision Processes (MDPs) often fail to provide identifiability guarantees, particularly for time-invariant latent confounders. For instance, Contextual MDPs [27] consider contextual variables that influence transition probabilities and rewards, but these variables are assumed to be partially observable, and their identifiability is not guaranteed. Factored Non-stationary MDPs [13] achieve identifiability

under specific structural assumptions but do not account for confounders that remain fixed over time. Other frameworks, such as Block MDPs [89] and POMDPs [62], and Latent MDPs [48], consider latent states or spaces but lack comprehensive modeling of how these latent factors influence state transitions and fail to provide theoretical guarantees for identifiability. Additionally, some works that do provide theoretical guarantees [36, 83], primarily focus on time-varying latent variables, leveraging advances in nonlinear independent component analysis (ICA) to establish identifiability. While these approaches are effective in their specific settings, the identifiability of time-invariant latent confounders remains underexplored, despite its practical importance in applications such as personalized decision-making and policy adaptation.

In contrast to these works, our method introduces a novel framework that explicitly models latent group factors influencing each state in the transition process. We provide theoretical guarantees of identifiability in both finite and infinite latent factor settings. Specifically, our approach ensures group-wise identifiability even under nonparametric transition processes and extends identifiability to infinite latent factors under appropriate assumptions. This establishes novel theoretical insights for learning state-transition processes with latent factors.

## F  Experiment Details

### F.1  Evaluation Metrics

**Pearson Correlation Coefficient**  Pearson Correlation Coefficient (PCC) [8] is a statistical measure that quantifies the degree of linear relationship between two variables. It provides a value between -1 and 1, where 1 implies a perfect positive linear relationship, -1 implies a perfect negative linear relationship, and 0 implies no linear relationship between the variables. The equation for calculating the Pearson Correlation Coefficient $r$ between two variables $X$ and $Y$ is as follows:

$$r = \frac{n(\sum xy) - (\sum x)(\sum y)}{\sqrt{[n \sum x^2 - (\sum x)^2][n \sum y^2 - (\sum y)^2]}} \tag{22}$$

where $n$ is the number of paired samples, $\sum xy$ is the sum of the product of paired scores, $\sum x$ and $\sum y$ are the sums of the $x$ scores and $y$ scores respectively, $\sum x^2$ and $\sum y^2$ are the sums of the squared $x$ scores and $y$ scores respectively.

**Canonical Correlation Analysis**  Canonical Correlation Analysis (CCA) [32] is designed to identify bases for two sets of variables in order to maximize the mutual correlations between the projections onto these bases. Let $X$ and $Y$ be the two sets of observed variables. This algorithm starts by centering the columns of $X$ and $Y$ so that they have zero mean. Then the covariance matrices $C_{XX} = X^\top X, C_{YY} = Y^\top Y$, and $C_{XY} = X^\top Y$ are calculated. After that, the canonical correlations are obtained by solving the following generalized eigenvalue problem: $C_{XX}^{-1} C_{XY} C_{YY}^{-1} C_{YX} \nu = \lambda \nu$. The square roots of the eigenvalues $\lambda$ indicate the canonical correlations between the linear combinations of $X$ and $Y$. The corresponding eigenvectors $\nu$ and $u = C_{XY}\nu$ are the canonical weights used to construct the canonical variables. Finally, the canonical variables of $X$ and $Y$ are $U = X\nu$ and $V = Yu$, respectively, representing the linear combinations of the original variables that are maximally correlated. The correlation of the primary pair of canonical variables is the highest, followed by the secondary pair, and so on. When employing CCA as an evaluation metric, a higher canonical correlation indicates a stronger and more relevant relationship between the recovered latent variable and the ground truth latent variable.

To extend the capability of CCA for analyzing nonlinear relationships, Kernel Canonical Correlation Analysis (KCCA) [81] is employed in the experiment which uses kernel functions to map the original variables into a higher-dimensional feature space. In our work, KCCA is used as an evaluation metric to validate that the recovered latent variable is meaningfully related to the ground truth latent variable, thus proving the relevance of the estimated representations. This allows for the capture of more complex, nonlinear correlations between the variables, thus potentially increasing the robustness and relevance of the relationships discovered in scenarios where linear methods fall short.

### F.2  Dataset Descriptions

**Synthetic Data Generation Processes**  In this paper, we created three synthetic datasets: Case 1 corresponds to a finite latent factor that satisfies our assumptions, and Case 2 and Case 3 allow

for multiple finite and infinite latent variables. The dimensions of states and actions are set to 3 and 2, respectively. The actions taken are generated randomly, following a uniform distribution $\text{Uniform}(0, 1)$. The noise term follows a mean-zero Gaussian distribution. The mixing function $f$ corresponds to the post-nonlinear model [90], where $f_1$ represents the nonlinear effect, and $f_2$ denotes the invertible post-nonlinear distortion on $\mathbf{s}_t$, embodied by a randomly initialized three-layer MLP with Tanh activation function. The data generation process follows:

$$\mathbf{s}_t = f_2(f_1(\mathbf{s}_{t-1}, \mathbf{a}_{t-1}, \kappa), \epsilon_t)). \tag{23}$$

**PersuasionForGood**  The PersuasionForGood dataset reveals the mechanics of persuasion in the context of charitable giving. It contains 1017 dialogues from 1285 participants in which one participant, called the persuader (ER), tries to convince the other participant, called the persuadee (EE), to donate to a charity. An example dialog is shown in Figure 11. All participants underwent personality assessments, which included detailed participant-level information such as demographics, Big Five personality traits, moral foundations, and so on, allowing for a multifaceted analysis of persuasion strategies and allowing us to use the labeled 32-dimensional personalities of each persuader as the ground-truth latent factor in our experiments.

We use this dataset to evaluate the performance of our estimation framework. We compare the estimated factors to the documented personality traits of persuaders. By examining the interactions between participants with different backgrounds and personalities, we aim to identify underlying patterns that could create effective persuasive agents. Specifically, we use BERT embeddings to generate a 768-dimensional feature vector for each dialog utterance. This process starts with tokenization, segmenting words into smaller units. BERT then processes these tokens to produce contextual embeddings.

| Speaker | Utterance | Extrovert | Agreeable | Conscientious | Neurotic | Open |
|---|---|---|---|---|---|---|
| ER | Hello. How are you? | 3.6 | 4.4 | 4.4 | 3 | 4 |
| EE | I'm good, how are you doing? | 4 | 5 | 4.2 | 3.6 | 4.8 |
| ER | Very well. I'm just up organizing info for my charity. Are you involved with charities? | 3.6 | 4.4 | 4.4 | 3 | 4 |
| EE | Yes! I work with children who have terminal illnesses. What charity are you involved in? | 4 | 5 | 4.2 | 3.6 | 4.8 |
| ER | That's great! I help with Save The Children. | 3.6 | 4.4 | 4.4 | 3 | 4 |
| EE | Amazing! Working with kids is the best. What do you do for Save the Children? | 4 | 5 | 4.2 | 3.6 | 4.8 |
| ER | I help raise donations and volunteer time. | 3.6 | 4.4 | 4.4 | 3 | 4 |
| EE | That's so important. How do you raise donations? | 4 | 5 | 4.2 | 3.6 | 4.8 |
| ER | By directly asking for aid. Do you currently donate to your charity? | 3.6 | 4.4 | 4.4 | 3 | 4 |
| EE | Yes I do, but I'm happy to donate to yours as well! | 4 | 5 | 4.2 | 3.6 | 4.8 |
| ER | Wonderful! Would you be will to donate $1.00 of your task money to help Save the Children? Save The Children is an international non-governmental organization that promotes children's rights, provides relief and helps support children in developing countries. | 3.6 | 4.4 | 4.4 | 3 | 4 |
| EE | Yes, I would be happy to! | 4 | 5 | 4.2 | 3.6 | 4.8 |
| ER | Would $2.00 be too much to ask? | 3.6 | 4.4 | 4.4 | 3 | 4 |
| EE | No, I can do it. | 4 | 5 | 4.2 | 3.6 | 4.8 |
| ER | Thank you. Can we make it $1.50? These children really need the assistance. | 3.6 | 4.4 | 4.4 | 3 | 4 |
| EE | $1.50 sounds good then. | 4 | 5 | 4.2 | 3.6 | 4.8 |
| ER | Why not $1.75 then? :-) | 3.6 | 4.4 | 4.4 | 3 | 4 |
| EE | I can do $2.00! Happy to help. | 4 | 5 | 4.2 | 3.6 | 4.8 |
| ER | Thank you so much! Do you have any more questions for me? | 3.6 | 4.4 | 4.4 | 3 | 4 |
| EE | Nope. Thank you! | 4 | 5 | 4.2 | 3.6 | 4.8 |

Figure 11: A sample persuasive dialog between persuader (ER) and persuadee (EE) from the PersuasionForGood corpus, along with the Big Five personality test scores, including Openness, Conscientiousness, Extroversion, Agreeableness, and Neuroticism.

**Pendulum**  The pendulum environment, provided by OpenAI Gym, is a classic control task used for the evaluation RL models. This environment presents a continuous control task where the agent must learn to control a frictionless pendulum with the goal of swinging it to the highest point and keeping it in the inverted position. The pendulum starts at a random position, and the goal is to bring it to a standstill at the inverted position with the least amount of effort. The system is characterized by a continuous action space, representing the torque applied to the pendulum's fulcrum. For a pendulum of length $l$ and mass $m$, subject to gravity $g$ and a control input $u$, the equations of motion can be described by the following second-order nonlinear ordinary differential equations $\dot{\theta} = \omega$, $\dot{\omega} = -\frac{g}{l}\sin(\theta) + \frac{u}{ml^2}$, where $\theta$ is the angle of the pendulum from the vertical upright position, and

$\omega$ is the angular velocity of the pendulum. The state of the pendulum at any time $t$ can be represented as $\vec{s}_t = [\cos\theta_t, \sin\theta_t, \omega_t]$, action represents the torque applied to the free end of the pendulum in the range $a_t \in [-2, 2]$, and the reward function is defined as: $r_t = -(\theta_t^2 + 0.1 * \omega_t^2 + 0.001 * a_t^2)$.

The goal of RL algorithms is to determine an optimal control policy $\pi^*$ that minimizes the effort to swing and balance the pendulum upright, typically by minimizing a cost function defined over states and actions. Each episode provides a continuous stream of observations, actions, and rewards, allowing the development and evaluation of algorithms capable of learning effective control policies in continuous action spaces. In academic studies, the Pendulum environment serves as a benchmark to investigate the effectiveness of RL algorithms in handling continuous control tasks.

**HeartPole**    HeartPole provides a straightforward scenario for assessing healthcare treatment, highlighting the complex interplay between productivity, health, and decision-making strategies. It simulates a professional's quest for increased productivity and examines the long-term health impacts of short-term choices, such as insufficient sleep, and intake of coffee and alcohol. The states include alertness, hypertension, intoxication, time since last sleep, total elapsed time, and total work done.

A productivity function and a heart attack risk function are defined over these variables, rewarding incremental productivity while imposing a significant penalty for heart attacks. Every thirty minutes, the agent evaluates the current state and chooses from a set of actions: work, drink coffee (which increases alertness and hypertension), drink alcohol (which decreases alertness while increasing hypertension and intoxication), or sleep (time-consuming but essential to reduce hypertension and intoxication to maintain alertness).

**Half Cheetah**    Half Cheetah is an integral part of the Mujoco physics engine, designed to simulate the agility and mechanics of a cheetah through a 2D robotic model. This model consists of 9 body parts, including a torso, two front and two back thighs, shins and feet, connected by 8 joints to provide fluid motion reminiscent of a cheetah's natural gait. The primary goal for this robot is to achieve maximum forward speed while maintaining stability, mirroring the efficiency and speed of its biological counterpart.

The observational data in this environment includes both the position and velocity of each segment of the half cheetah, methodically ordered with all position data provided before velocity information. This systematic ordering allows for a detailed understanding of the dynamics of the robot at any given time. Actions within this simulation are defined by the torque applied to the joints, which directly affects its acceleration and motion patterns. The reward function for Half Cheetah is designed to encourage rapid forward motion and operational efficiency and consists of two main components: a forward motion reward proportional to the increase in the robot's horizontal displacement over time and a control cost penalty for unnecessary control effort and external force application. This reward structure is carefully designed to encourage the optimization of forward motion, with an emphasis on reducing control effort and mitigating forces that may interfere with the robot's streamlined motion.

## F.3    Additional Experiment Results

**Ablation Study: Variability in number of samples**    Here, we vary the number of samples to further verify this effectiveness. The data generation process is the same as Case 1, except that we change the number of samples to $\{100, 150, 200, 300, 500, 800, 1000\}$. The comparison results shown in Figure 14 indicate that our method can achieve consistently good recovery performance under different numbers of individuals. This further confirms that the identifiability of our framework is guaranteed by the mathematical relationship between the trajectory length and the number of groups, which is constrained by the sample sufficiency assumption under the conditions given in Theorem 4.1. Moreover, Figures 14(b) and 14(c) show the successful recovery of the latent group factor, validated by high-frequency similarity and a remarkable PCC value, confirming the ability of our method to skillfully recover latent variables in practical pendulum tasks.

**Ablation Study: Variability in initial states**    We conduct additional experiments to verify how the variability in initial states across individuals affects performance. The initial state distributions are defined with two types: normal and uniform. For the normal distribution, the means are set to [0, 1, 1] and the standard deviations are set to [1, 2, 1], respectively. For the uniform distribution, the range for each dimension is defined with lower bounds [0, -1, 1] and upper bounds [1, 1, 1.5]. The experiment

results in Figure 15 show that although the initial states have high variability, the estimated values of the latent factors corresponding to the 200 individuals are ultimately highly classified and can be divided into 4 groups.

**Ablation Study: Consideration of transformer as encoder**   Our framework is flexible enough to integrate various encoders and decoders, depending on the application tasks. To demonstrate this flexibility, we incorporated Transformers into our framework and conducted a comparative analysis against the existing models. The result, shown in Figure 15(c), indicates that while both frameworks achieve identifiability, the Transformer-based encoder demonstrates faster convergence compared to our previous approach.

**Added Experiment: Inventory**   Inventory management [69] is an important real-world problem that aims to keep inventories of goods at optimal levels to minimize inventory costs while maximizing revenue from demand fulfillment. We tested the performance of our algorithm on the inventory with state dimensions of 50, 100, and 200 and added additional baselines (8) Meta gradient RL, (9) Multitask RL, (10) Policy distillation, and (11) Non-policy adaptation to verify the model. The experimental results in Figure 12 show that our framework outperforms other algorithms in terms of initial reward and final reward.

**Added Experiment: AhnChemo**   AhnChemoEnv [59] is designed to simulate cancer treatment through chemotherapy, allowing realistic modeling of tumor growth and response to treatment. We create different groups with PK/PD variation. The experimental results in Figure 13 show that our framework outperforms other algorithms in terms of initial and final reward. Our method achieves the highest initial and final rewards compared to the baselines. Specifically, it shows a significant jump-start compared to non-policy adaptation, validating the effectiveness of our adaptation approach.

The meta-gradient method optimizes the hyperparameters of the learning algorithm by calculating the gradient of the learning process, allowing rapid adaptation to new tasks as they change. However, due to the continuous adjustment of learning strategies during training, it converges more slowly and the adaptation effect is less significant compared to our algorithm. Multitask RL improves learning efficiency by sharing model strategies across different tasks. This requires first training policies on multiple tasks, which can be time-consuming (and even risky) during exploration. Moreover, identifying which new task corresponds to a previously trained task can be challenging. Our algorithm addresses this by estimating directly without requiring prior knowledge. Policy distillation transfers the knowledge of already trained teacher models to a student model, allowing the student to perform well across multiple tasks. However, this approach highly relies on the performance of the teacher models; insufficiently trained teacher models can negatively impact the final performance. Our algorithm does not depend on the source policy performance; subsequent policy optimization is based on the new environment, leading to better final performance.

### F.4   Training Details

The estimation framework is trained using AdamW optimizer for a maximum of 200 epochs and early stops if the validation ELBO loss does not decrease for ten epochs. A learning rate of 0.001 and a mini-batch size of 32 are used. We used three random seeds in each experiment and reported the mean performance with standard deviation averaged across random seeds. We used a machine with the following CPU specifications: 11th Gen Intel(R) Core(TM) i7-11800H @ 2.30GHz with 16 logical processors. The machine has one GeForce RTX 3080 GPU with 32GB GPU memory.

## G   Impact Statement

Our work has a significant impact on ethics, society, and future applications. We emphasize the importance of individualized policies in systems and advocate for a deeper understanding and respect for individual differences. Tailoring interventions to different individuals has the potential to improve user experience and outcomes in healthcare, education, and other areas. This approach avoids a one-size-fits-all policies. Our method can greatly improve individualized services, transforming the delivery of educational content, the management of healthcare, and the recommendation of products. This makes these services more effective and aligned with individual needs. However, the implementation of this method requires careful consideration of privacy and data security, as

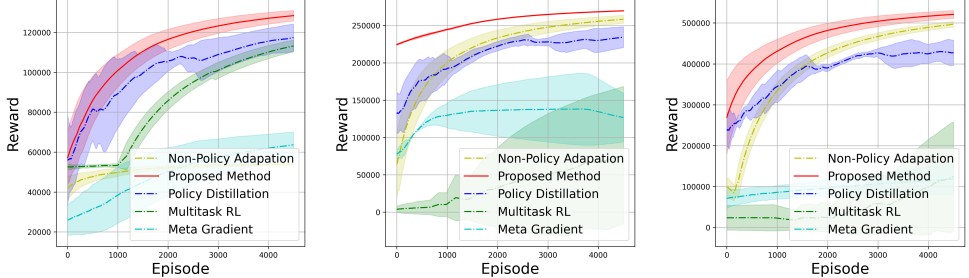

(a) Reward curve with $d_s = 50$.  (b) Reward curve with $d_s = 100$. (c) Reward curve with $d_s = 200$.

Figure 12: **Results in Inventory**. We evaluated the performance of different methods under different state dimensions. Our algorithm scales well to high-dimensional cases and outperforms other baselines in terms of initial reward and final reward.

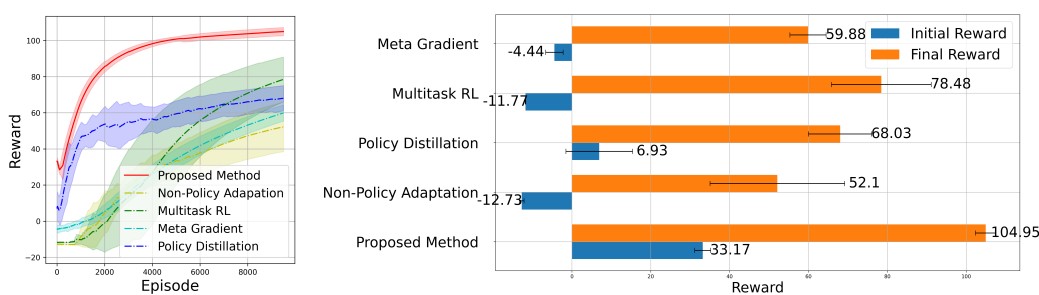

(a) Accumulative reward curves.    (b) Initial and final reward under different benchmarks.

Figure 13: **Results in AhnChemoEnv**. We evaluated the performance of our method against several baselines, including (1) meta gradient RL, (2) multitask RL, (3) policy distillation, and (4) non-policy adaptation. Our method outperforms these benchmarks and achieves superior performance in terms of initial reward and final reward.

personalized systems require the collection and analysis of personal data. Maintaining user trust, preventing misuse, and ensuring ethical use of such data are of utmost importance.

## H  Estimation Framework Details

The proposed framework is customized based on the requirements of the identifiability theorems given in Section 4. We would like to emphasize that our proposed framework differs from the traditional VAE and model-based RL in three main aspects: (1) Our framework uses a quantization layer to discretize the continuous latent representations. This mapping of continuous latent representations to an embedding dictionary is well suited to the group determinacy requirement. (2) Our decoder reconstructs individualized state transition processes to simulate the data generation process, incorporating additional conditions as well as the estimated latent factor. (3) We further extract latent factors for each individual as additional information to facilitate individual policy learning. The detailed implementations of each component are summarized below.

**Encoder**    For any individual $m$, the Conv1D layer transforms an input sequence $\mathbf{s}_t^m$, using learned kernel filters. These filters slide over the sequence to produce a feature map, denoting the response of the filter at each position. Mathematically, the transformation by a single filter in the Conv1D layer at time $t$ is described as $\mathbf{o}_t = \sigma\left(W * \mathbf{s}_{t:H+t}^m + b\right)$, where $\mathbf{o}_t$ is the feature map, $W$ the kernel to be learned during training, $*$ the convolution operation, $\mathbf{s}_{t:H+t}^m$ the input sub-sequence from time $t$ to $t + H$, where $H$ is the size of the kernel. $\sigma$ is the activation function, and $b$ is the bias term to be learned during training. The layer may contain multiple such filters, each learning different features

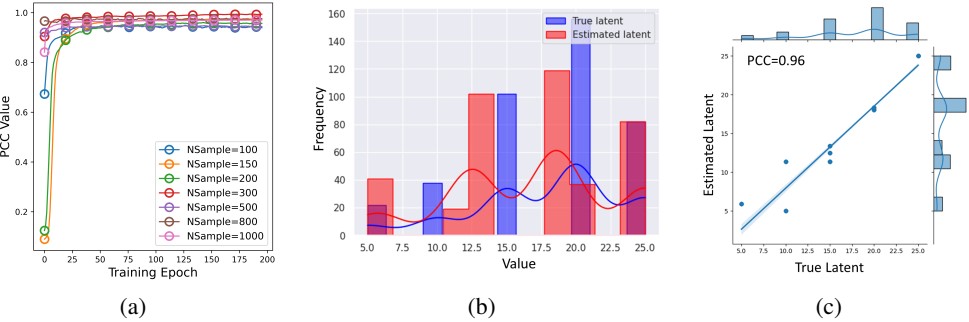

(a)           (b)           (c)

Figure 14: (a) PCC trajectory comparisons under different numbers of individuals. (b-c) Successful recovery of the latent group factor in the Pendulum.

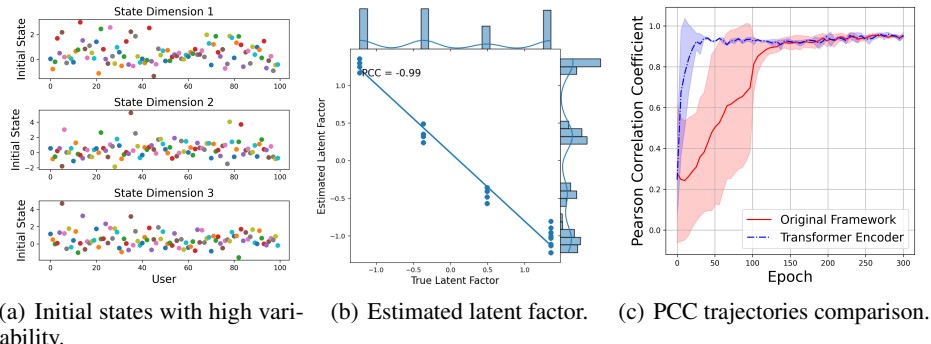

(a) Initial states with high variability.    (b) Estimated latent factor.    (c) PCC trajectories comparison.

Figure 15: (a-b) **Evaluation on variability in initial states**. The estimated values of $\kappa$ are highly clustered into four classes. (c) **Incorporating Transformers**. The transformer encoder achieves faster convergence compared to the original framework.

of the input sequence. The resulting feature maps serve as a transformed representation $z_m$, which embeds the information about the latent group factor $\kappa$.

As for the LSTM, let the hidden states and cell states of the LSTM at time $t$ denote as $h_t$ and $c_t$, respectively. The final hidden state of LSTM $h_T$, after the sequential processing of the entire trajectory, serves as the representative $z_m$ that embeds the information about the latent group factor $\kappa$.

**Quantization Layer**    Let the output of the encoder be a continuous latent representation denoted as $z_m \in \mathbb{R}$, and define an embedding dictionary $E$ consisting of $G$ vectors, where each vector represents a unique discrete category: $E = \{e_1, e_2, \ldots, e_G\}$, where $e_i \in \mathbb{R}$. The quantized vector $\hat{\kappa}_m$ is obtained by mapping $z_m$ to the nearest dictionary vector. The mapping can be expressed mathematically as $\hat{\kappa}_m = \arg\min_{e_i \in E} \|z_m - e_i\|_2$. Subsequently, the quantized output is the vector from the dictionary that is closest to the encoder output. Thus, the continuous representation $z_m$ is effectively mapped to a discrete $\hat{\kappa}_m$ by finding the nearest neighbor in the dictionary, aligning the representation learning with the discrete nature of the latent variable.

**Decoder**    Suppose $\mathbf{s}_{t-1}^m$ and $\mathbf{a}_{t-1}^m$ as the true previous state and action, respectively. Let $\hat{\kappa}_m$ be the approximated latent group factor for the $m$-th individual. The inputs to the conditional decoder are a combination of the aforementioned variables: $\text{Input}_t = (\mathbf{s}_{t-1}, \mathbf{a}_{t-1}, \hat{\kappa}_m)$. The output of the decoder is the reconstructed next state, $\hat{\mathbf{s}}_t$, which is a function of the decoder input: $\hat{\mathbf{s}}_t = \text{De}(\text{Input}_t)$. The reconstruction likelihood measures how closely the reconstructed state matches the true subsequent state, which is defined as $\mathcal{L}_{\text{Recon}} = p_{\text{Recon}}(\mathbf{s}_t^m | \mathbf{s}_{t-1}^m, \mathbf{a}_{t-1}^m, \hat{\kappa}_m)$. The objective in this process is to optimize the decoder parameters to maximize the reconstruction likelihood $\max \mathcal{L}_{\text{Recon}}$ so that the reconstructed state $\hat{\mathbf{s}}_t$ is as close as possible to the true next state $\mathbf{s}_t$.

# I   Algorithm

The pseudocode for the proposed algorithm is presented in Algorithm 1 and Algorithm 2.

---

**Algorithm 1** Algorithm of Individualized Policy.

---

1: **Input:** $\{f_{\text{Env}}^m\}_{m=1}^M$: individualized environments; Encoder: encoder; Quantization: embedding dictionary; Decoder: decoder; $\pi$: policy network
2: **Output:** $\{\hat{\kappa}_g\}_{g=1}^G$: estimated latent factor; $\{\pi_m^*\}_{m=1}^M$: optimized individualized policy
3:
4: *## Main loop*
5: **Main**($f_{\text{Env}}$, Encoder, Quantization, Decoder, $\pi$)
6: Encoder, Quantization, Decoder, $\pi \sim \text{N}(0, \text{I})$ *# Randomly initialize the network*
7: $\mathcal{H} \leftarrow \{\tau_m\}_{m=1}^M$ *# Collect individual trajectories by interaction with $\{f_{\text{Env}}^m\}_{m=1}^M$*
8: **for** each individual $m$ **do**
9:     $z_m = \text{Encoder}(\mathbf{s}_{0:T}^m)$ *# Capture the high-level representations*
10:    $\hat{\kappa}_m = \text{Quantization}(z_m)$ *# Vector quantization*
11:    **for** each state $\mathbf{s}_t^m$ in the trajectory **do**
12:        $\hat{\mathbf{s}}_t^m = \text{Decoder}(\mathbf{s}_{t-1}^m, \mathbf{a}_{t-1}^m, \hat{\kappa}_m)$ *# Reconstruct the next state*
13:    **end for**
14: **end for**
15: **return** $\{\pi_g^*\}_{g=1}^G = \text{PolicyLearning}(\mathcal{H}, \{\hat{\kappa}_g\}_{g=1}^G)$ *# Optimize the individualized policies*
16:
17: **EncoderFunction**($\mathbf{s}_{0:T}^m$)
18: **if** encoder is Conv1D **then**
19:    **for** each $t$ in $\mathbf{s}_{0:T}^m$ **do**
20:        $o_t^m \leftarrow \text{Conv1D}(\mathbf{s}_{t:t+H}^m)$
21:    **end for**
22: **else if** encoder is LSTM **then**
23:    Initialize $h_0^m, c_0^m$
24:    **for** each $t$ in $\mathbf{s}_{0:T}^m$ **do**
25:        $h_t^m, c_t^m \leftarrow \text{LSTM}(h_{t-1}^m, c_{t-1}^m, \mathbf{s}_t^m; \theta)$
26:    **end for**
27: **end if**
28: **return** $z_m \leftarrow$ Final output of Conv1D or final hidden state of LSTM
29:
30: **QuantizationFunction**($z_m$)
31: Initialize $E = \{e_1, e_2, \ldots, e_G\}, d_{\min} = \infty$
32: **for** each $e_i$ in $E$ **do**
33:    **if** $\|z_m - e_i\|_2 < d_{\min}$ **then**
34:        Update $d_{\min}$ and $\hat{\kappa}_m \leftarrow e_i$
35:    **end if**
36: **end for**
37: **return** $\hat{\kappa}_m$
38:
39: **DecoderFunction**($\mathbf{s}_{t-1}^m, \mathbf{a}_{t-1}^m, \hat{\kappa}_m$)
40: Reconstruct state based on condition $\hat{\mathbf{s}}_t^m \leftarrow \text{Decoder}(\mathbf{s}_{t-1}^m, \mathbf{a}_{t-1}^m, \hat{\kappa}_m)$
41: **return** Reconstructed state $\hat{\mathbf{s}}_t^m$
42:
43: **PolicyLearningFunction**($\mathcal{H}, \{\hat{\kappa}_g\}_{g=1}^G$)
44: **for** each individual $m$ **do**
45:    Update policy input to $\mu_\pi(\mathbf{s}_t; \theta^\mu) \rightarrow \mu_\pi^m(\mathbf{s}_t^m, \hat{\kappa}^m; \theta^\mu)$
46:    Update training objective:
47:        $\mathcal{J}(\theta^\mu) = \mathbb{E}\left[\sum_{t=0}^\infty \gamma^t Q\left(\mathbf{s}_t, \mu_\pi^m(\mathbf{s}_t^m, \hat{\kappa}_m; \theta^\mu); \theta^Q\right)\right]$
48:    Optimize $\mu_\pi^m$ for individual $m$
49: **end for**
50: **return** Optimized individual policy $\mu_\pi^*$

---

---

**Algorithm 2** Training Process with Augmented ELBO Objective.

---

1: Initialize parameters of the Encoder and Decoder
2: Initialize weights $\alpha$ and $\beta$
3: **repeat**
4:   **for** each individual $m$ **do**
5:     Compute encoded representation: $z_m \leftarrow \text{Encoder}(\mathbf{s}_{0:T}^m)$
6:     Estimate individual-specific factor: $\hat{\kappa}_m \leftarrow \text{Quantization}(z_m)$
7:     Compute reconstructed state: $\hat{\mathbf{s}}_t^m \leftarrow \text{Decoder}(\mathbf{s}_{t-1}^m, \mathbf{a}_{t-1}^m, \hat{\kappa}_m)$
8:     Calculate $\mathcal{L}_{\text{Recon}} = \sum_t \|\mathbf{s}_t^m - \hat{\mathbf{s}}_t^m\|^2$
9:     Calculate $\mathcal{L}_{\text{Quant}} = \sum_i \|\text{sg}[z_{m,i}] - e_{m,i}\|^2$, $\mathcal{L}_{\text{Commit}} = \sum_i \|z_{m,i} - \text{sg}[e_{m,i}]\|^2$
10:    Compute extended ELBO objective: $\mathcal{L}_{\text{ELBO}} = \mathcal{L}_{\text{Recon}} + \alpha \mathcal{L}_{\text{Quant}} + \beta \mathcal{L}_{\text{Commit}}$
11:    Update parameters to optimize $\mathcal{L}_{\text{ELBO}}$
12:   **end for**
13: **until** convergence

---

