# OpenReview forum: "Identifying Latent State-Transition Processes for Individualized Reinforcement Learning"
_NeurIPS.cc/2024/Conference — NeurIPS 2024 poster_

### Official Review · Reviewer_pu6U · 2024-07-12

**Soundness:** 3
**Presentation:** 4
**Contribution:** 3
**Rating:** 5
**Confidence:** 3

**Summary:**

The paper addresses the challenge of identifying individualized state-transition processes in reinforcement learning (RL) when individual-specific factors are latent. The authors propose a novel framework, the Individualized Markov Decision Processes (iMDPs), to incorporate these latent factors into the state-transition processes. They provide theoretical guarantees for the identifiability of these latent factors and present a practical method for learning them from observed state-action trajectories. Experiments on various datasets demonstrate the effectiveness of their method in identifying latent state-transition processes and optimizing individualized RL policies.

**Strengths:**

1.  Theoretical guarantees for identifiability under various conditions.
2.  Effective practical method for learning individualized RL policies from observed data.
3. Comprehensive experiments on synthetic and real-world datasets demonstrating the effectiveness of the proposed method.

**Weaknesses:**

1. The method might be challenging to generalize to all types of RL problems, especially those with instantaneous causal influences within states.
2. Some sections could be simplified for broader accessibility, and future work should address the limitations mentioned.

**Questions:**

1.How does the proposed framework scale with larger datasets and more complex environments? Are there any computational limitations or bottlenecks that need to be addressed?
2. Can the framework handle real-time adaptation in dynamic environments where latent factors might change over time? If so, how would this be implemented?

**Limitations:**

1. The framework assumes that the latent individual-specific factors are time-invariant. In real-world scenarios, individual-specific factors can evolve over time, which this model does not account for.
2. The framework might face challenges in scaling to environments with very high-dimensional state and action spaces, which are common in some real-world applications.

---

> ### Author Rebuttal · Authors · 2024-08-07
>
> **Q1: The method might be challenging to generalize to all types of RL problems, especially those with instantaneous causal influences within states.**
>
> **A1:** Thank you for pointing out this scenario. It is indeed possible that there are instantaneous causal influences within states in the considered problem. However, even with these influences, our framework can still identify the latent individual-specific factors, and the theoretical results will not be affected. In the instantaneous case, since the states are observed, we can treat the states at each time step as a whole and input them into the current estimation framework. Thus the instantaneous causal influences within states will not affect our current estimation framework and theorem results.
>
> **Q2: Some sections could be simplified for broader accessibility, and future work should address the limitations mentioned.**
>
> **A2:** Thank you for your suggestion. In the updated manuscript, we have reorganized the sections to improve accessibility and clarity. Additionally, we have added a separate Limitations section to discuss how we address limitations, including instantaneous causal influences.
>
> Future work could extend our current framework from a causal perspective. For instance, we could develop modules that explicitly account for causal dependencies within states, using techniques such as causal graphical models and advanced inference methods that handle instantaneous causal relationships [1]. Such extensions could be directly integrated into our current framework, and we have been working on this problem.
>
> [1] Li, Zijian, et al. "On the Identification of Temporally Causal Representation with Instantaneous Dependence." arXiv preprint arXiv:2405.15325 (2024).
>
> **Q3: How does the proposed framework scale with larger datasets and more complex environments? Are there any computational limitations or bottlenecks that need to be addressed? The framework might face challenges in scaling to environments with very high-dimensional state and action spaces, which are common in some real-world applications.**
>
> **A3:** In the updated manuscript, we validated our algorithm in the AhnChemoEnv in DTRGym [2] and inventory management tasks [3]. AhnChemoEnv is designed to simulate cancer treatment through chemotherapy, allowing realistic modeling of tumor growth and response to treatment. Inventory management is an important real-world problem that aims to keep inventories of goods at optimal levels to minimize inventory costs while maximizing revenue from demand fulfillment. We test the performance of our algorithm on the inventory with state dimensions of 50, 100, and 200. The experimental results (see Figure 1 and Figure 3 in REBUTTAL.pdf) show that our framework outperforms other algorithms in terms of initial reward and final reward.
>
> Specifically, our method shows a significant jump-start compared to non-policy adaptation, validating the effectiveness of our approach. Unlike the meta-gradient method, which requires continuous adjustment of learning strategies during training and thus converges more slowly with less significant adaptation effects, our algorithm is more efficient. Additionally, training across multiple tasks in multitask RL can be time-consuming, and identifying which new task corresponds to a previously trained task can be challenging. Our algorithm addresses this by directly estimating $\kappa$ without requiring prior knowledge. Furthermore, policy distillation heavily depends on the performance of teacher models; insufficiently trained teacher models can negatively impact the final performance. Our algorithm, however, does not rely on source policy performance. Instead, it optimizes the policy based on the new environment, leading to better final performance.
>
> Our findings suggest that while the algorithm scales reasonably well, certain computational limitations, such as increased processing time and memory consumption, need to be addressed. In high-dimensional scenarios, the use of hardware accelerators such as GPUs, as well as fine-tuning the model, is essential to maintain optimal performance.
>
> [2] Luo, Zhiyao, et al. "DTR-Bench: An in silico Environment and Benchmark Platform for Reinforcement Learning Based Dynamic Treatment Regime." arXiv preprint arXiv:2405.18610 (2024).
>
> [3] Sun, Yuewen, et al. "ACAMDA: Improving Data Efficiency in Reinforcement Learning Through Guided Counterfactual Data Augmentation." Proceedings of the AAAI Conference on Artificial Intelligence. Vol. 38. No. 14. 2024.
>
> **Q4: Can the framework handle real-time adaptation in dynamic environments where latent factors might change over time? If so, how would this be implemented? The framework assumes that the latent individual-specific factors are time-invariant. In real-world scenarios, individual-specific factors can evolve over time, which this model does not account for.**
>
> **A4:** Thank you for pointing out this interesting scenario. It is indeed possible for the latent individual-specific factor to be time-variant in the considered problem. We believe our framework can be extended to handle time-varying cases, although establishing the theoretical identifiability is highly non-trivial. For instance, if we directly allow these latent factors to be time-varying, it seems hopeless to recover the latent variables, since each individual may have a specific latent factor. Therefore, further constraints would be needed. For instance, the theoretical identifiability might benefit from the constraint that the total number of possible values of the latent factors over time and across different individuals is finite. We hope this important extension can be achieved by researchers in the field by leveraging our framework and further considering the idea in [4].
>
> [4] Hu, Yingyao, and Matthew Shum. "Nonparametric identification of dynamic models with unobserved state variables." Journal of Econometrics 171.1 (2012): 32-44.

---

> ### Author Response · Authors · 2024-08-12
> **Could you please let us know whether our responses properly addressed your concern?**
>
> Dear Reviewer pu6U,
>
> Thank you again for your valuable comments. Your suggestions on the experiments were very helpful, and we are eager to learn whether our additional experimental results and our responses have properly addressed your concerns.
>
> Due to the limited time for discussion, we hope to see your feedback and hope for the opportunity to respond to your further questions, if there are any.
>
> Yours sincerely,
>
> Authors of submission 12027

---

### Official Review · Reviewer_t6CB · 2024-07-12

**Soundness:** 2
**Presentation:** 3
**Contribution:** 3
**Rating:** 6
**Confidence:** 3

**Summary:**

The authors consider a problem where the underlying dynamical process is an MDP but with a time-invariant latent factor. The authors provide examples of such problems in the real-world. To solve these problems, the authors propose a new mathematical framework called Individualized Markov Decision Processes (iMDPs). Furthermore, the authors provide theoretical guarantees and an algorithm for learning these latent factors from observational data. The authors then demonstrate the performance of this approach on multiple datasets.

**Strengths:**

I think the problem considered is a very relevant and useful problem. The authors propose a novel approach to solve this problem and present theoretical guarantees and analyses along with some empirical experiments to validate their approach. The paper is, for the most part, well written and clear.

**Weaknesses:**

The authors could have considered simultaneous estimation and control, which is often needed in real-world problems. Furthermore, in choosing baselines for comparisons, the authors have not considered RL methods that take into account trajectory history.

**Questions:**

1. How does this compare with meta learning or transfer learning or a POMDP where each individual is a "slightly different" environment? While the authors state that these models are different, in terms of learning and practical considerations, how do these frameworks differ?
2. Does identifiability of these latent factors get influenced by the policy being followed to generate the data?
3. Why is the individual-specific latent factor $\kappa$ not included in the reward function?
4. What does the term $\mathbb{P}(s, a | u)$ imply in equation (1)? Since $u$ is not observed by the decision maker, how can its action $a$ be conditioned on $u$?
5. $d_s$ and $d_a$ are not defined in line 110. Can the authors add these definitions?
6. Is this learning online or offline? From Line 246 it appears that the authors only consider the offline case. However, their control part requires an online setup as mentioned in Line 335. Specifically is the learning of $\kappa$ and optimal policy simultaneous?
7. If an online setup was available, how would the authors propose to use exploration to better identify $\kappa$? Will this exploration to learn $\kappa$ be harmful/costly?
8. Can the authors define the term "post-nonlinear temporal" model in Line 162?
9. Can the authors define the covariance matrices $\Sigma_{\mathbf{A_i}, \mathbf{B_i}}$ in Line 173?
10. Is $n$ in Theorem 4.3 same as $d_s$ in Line 110?
11. Is the iMDP assumed to be an infinite horizon setup?
12. Why are the quantization loss and commitment loss defined after equation (4) the same? Also in Line 9 in Algorithm 2.
13. Is the reconstruction loss differentiable? Can the authors explain as it involves a quantization step as well? Or does this term only compute the gradient with respect to parameters of the decoder?
14. Is the policy adaptation step specified equivalent to augmenting the input given to the policy function or Q function with $\kappa$?
15. In the caption for Figure 3, it would be helpful if the authors could define the shaded regions around the curves.
16. While considering RL baselines, why haven't algorithms for POMDPs that take the entire history into account been considered?
17. Can the authors explain the motivation behind the ablation study starting in Line 317?
18. What do the authors mean by the limitation about "instantaneous causal influences" in Line 389?
19. Can the authors give more details in the proof of Theorem 4.2? (Line 678 onwards)
20. In Line 678, is $n$ the number of state components?
21. I could not follow the proof for Theorem 4.3. There are several concepts introduced in the appendix and often the proof refers to a concept explained later. It would be helpful if the authors could provide some motivation/intuition for these various constructions/concepts.
22. In line 912, arg max is a set operator and hence $=$ should be replaced with $\in$.
23. In Line 7 in Algorithm 1, what is the policy used to collect the trajectory samples?
24. Can the authors give the specific values for the dimensions and other architecture details used in their numerical examples?

**Limitations:**

The authors discuss the limitations of their work in the Conclusions section and in the Impact Statement section (Appendix H).

---

> ### Author Rebuttal · Authors · 2024-08-07
>
> **A1:** Thanks for the helpful questions. Here is a brief summary of the differences. We also empirically compared our method with meta-learning and transfer-learning techniques and reported results in REBUTTAL.pdf.
>
> - **Meta-learning** trains the model on a variety of tasks so that it can efficiently apply what it has learned to new tasks. Unlike our method, it does not assume a time-invariant latent factor, has no guarantee of identifiability, and does not provide a clear clue of adaptation.
>
> - **Transfer learning** focuses on leveraging knowledge from one domain to improve performance in a new domain while facing challenges such as negative transfer. Our method can identify the latent individual-specific factors and realize individualized decision-making with interpretability.
>
> - **POMDPs** focus on dealing with incomplete information and making decisions under uncertainty with time-varying hidden states. Our work emphasizes individualized decision-making considering latent factors that influence state transitions with fully observable states.
>
> **A2:** In the current theoretical treatment, yes, the identifiability is influenced by the policy. According to our proof technique, this assumption is needed. Thank you for pointing it out and we have made this condition explicit in the updated paper. At the same time, it might be possible to avoid this assumption although the proof seems to be nontrivial.
>
> **A3:** In our setting, we assume that different groups have no different reward weightings. Thus $\kappa$ only influences the states and transition dynamics.
>
> **A4:** We believe this might be a misunderstanding. $\mathbb{P}(s,a|u)$ represents the joint distribution of $(s,a)$ for each individual. Here, $u$ denotes different individuals which is explicitly observed. This notation is used for mathematical convenience.
>
> **A5, A10, A20:** Both $n$ in Theorem 4.3 and $d_s$ in line 110 represent the state dimension, and $d_a$ denotes the action dimension. We didn’t find $n$ in line 678–perhaps you referred to another line number? If you mean the $n$ in lines 166, 728, or 739, it also represents the state dimension.
>
> **A6, A7:** The estimation network is pre-trained offline. When a new individual arrives, we estimate $\kappa$ and adapt the policy simultaneously, and the policy is adapted through new interactions. Such exploration is necessary to better identify $\kappa$ and discover the optimal policy in RL. This has been provided explicitly in the updated paper.
>
> **A8, A9:** The definition of the post-nonlinear temporal model and covariance matrices are provided in Appendix C.1 and C.4.1, respectively. In light of your question, we have also moved them to the main paper in the updated paper.
>
> **A11:** Yes, we assume an infinite horizon setup in iMDP.
>
> **A12, A13:** This might be a misunderstanding. The quantization loss and the commitment loss are different and serve unique purposes. The key is in the positioning of the stop gradient (sg[·]) [1], showing how these losses influence the encoder and the codebook differently. The stop gradient allows the model to handle the non-differentiable quantization step, ensuring that gradients are computed for the encoder and the rest of the model, making the reconstruction loss differentiable.
>
> [1] Van Den Oord, Aaron, and Oriol Vinyals. "Neural discrete representation learning." Advances in neural information processing systems 30 (2017).
>
> **A14:** Yes, we identify and incorporate $\kappa$ into policy function to learn individualized policy.
>
> **A15:** The shaded regions represent the standard deviation. We have added an explanation in the revision.
>
> **A16:** Our problem and POMDP address different settings. We assume fully observable states influenced by latent individual-specific factors, rather than partially observable states. Under the similar setting, we compared our method with meta-learning and transfer learning techniques.
>
> **A17:** The ablation study analyzes the contributions of different components in the framework. The sequential encoder aligns our framework with the theorem requirements, and the conditional decoder meets our data generation process. The noise estimator is used to increase robustness.
>
> **A18:** Our current framework only considers transitions from time $t$ to $t+1$. However, in some cases, there may be causal influences that happen instantaneously within the same time step, which is beyond the scope of our current discussion.
>
> **A19:** Individuals sharing the same latent factor can be grouped due to similar state transitions. For a specific state-action pair $(s^*, a^*)$, the next state $s'$ should be consistent across individuals in the same group. We define $t^j$ as the time of the $j$-th occurrence of $(s^*, a^*)$ and collect states $X^j$ = {$s_{t^j+1} \mid t^j = 1, 2, \dots$}. By comparing $X^j$ across different individuals, we group those with similar transition behaviors and achieve identifiability by Lemma B.1.
>
> **A21:** We appreciate your feedback–indeed, this part is dense and involves many concepts and derivations. The proof for Theorem 4.3 has two parts.
>
> - Structure Identifiability: Lemma C.1 shows that the rank of the sub-covariance matrix of the observed variables equals the number of minimal choke sets, which represent the minimal sets that separate the two observed variable sets forming the sub-covariance matrix. The difference between the calculated rank and the expected choke factors indicates the number of latent factors.
>
> - Parameter Identifiability: We estimate $\alpha$ and $\beta$, which describe the influence of states and actions, using a regression model. The latent coefficient $\lambda$ is identified using orthogonal techniques in factor analysis.
>
> **A22:** We have revised the notation accordingly.
>
> **A23:** The trajectory samples are collected using a random policy.
>
> **A24:** The summary of architecture details is provided in REBUTTAL.pdf and more details have been added in the updated manuscript.

---

> ### Author Response · Authors · 2024-08-12
> **Could you please let us know whether our responses properly addressed your concern?**
>
> Dear Reviewer t6CB,
>
> Thank you again for your valuable time dedicated to reviewing our paper and for your helpful suggestions. We particularly appreciate your questions regarding the problem setting, theorem, and framework. So we are eager to see whether our responses properly addressed your concerns and would be grateful for your feedback.
>
> With best wishes,
>
> Authors of submission 12027

---

> > ### Comment · Reviewer_t6CB · 2024-08-12
> >
> > I thank the authors for their detailed explanations to my questions. Based on these responses, I have increased my score. However, there are some points that I have not yet understood:
> >
> > A. "POMDPs focus on dealing with incomplete information and making decisions under uncertainty with time-varying hidden states. Our work emphasizes individualized decision-making considering latent factors that influence state transitions with fully observable states". I did not understand this. If the states are fully observable, then in an MDP framework, the transitions should not depend on any other factor. Am I missing something here?
> >
> > B. Regarding A12 and A13, I find the definitions:
> > 1. $\mathcal{L}_{Quant} = \sum_i \lVert sq[z_i] -  e_i \rVert^2$
> > 2. $\mathcal{L}_{Commit} = \sum_i \lVert e_i - sq[z_i] \rVert^2$
> > to be identical (I have left out the $m$ in the subscript on the RHS due to a formatting error. What am I missing here?

---

> > > ### Author Response · Authors · 2024-08-13
> > >
> > > Thank you so much for your time and feedback. We sincerely appreciate you updating your recommendation. Please let us answer your questions below.
> > >
> > > **Follow-Up Q1: If the states are fully observable, then in an MDP framework, the transitions should not depend on any other factor. Am I missing something here?**
> > >
> > > **Follow-Up A1**: Thanks for the good question. According to the definitions [1-2], the difference between POMDPs and MDPs is that in POMDP, the agent is unable to directly observe the current state $s$. Instead, the POMDP agent only receives a noisy or partial observation $o$, which is decided by the sensor model $O(o|s, a)$ (or sometimes $O(o|s)$).
> > >
> > > In our problem setting, although the states $s$ are fully observable, the transition probability $\mathbb{P}(s’|s, a, \kappa)$ is influenced by unobserved variable $\kappa$, along with the observed variables ($s$ and $a$). The dependency of the transition on observed or unobserved variables does not necessarily determine whether the environment is modeled as an MDP or a POMDP. Therefore, despite including a latent variable, our setting conforms to the definition of an MDP.
> > >
> > > [1] Kaelbling, Leslie Pack, Michael L. Littman, and Anthony R. Cassandra. "Planning and acting in partially observable stochastic domains." Artificial intelligence 101.1-2 (1998): 99-134.
> > >
> > > [2] Igl, Maximilian, et al. "Deep variational reinforcement learning for POMDPs." International conference on machine learning. PMLR, 2018.
> > >
> > >
> > > **Follow-Up Q2: Regarding A12 and A13, I find the definitions to be identical. What am I missing here?**
> > >
> > > **Follow-Up A2**: Sorry for the confusion and you are right. Thank you for pointing out this typo, and we have corrected it in the updated manuscript. The quantization loss and commitment loss should be
> > >
> > > $\mathcal{L} _ {\text{Quant}} = \sum_i \left\|\| \text{sg}[z _ {m,i}] - e _ {m,i} \right\|\|^2$, and
> > >
> > > $\mathcal{L} _ {\text{Commit}} = \sum_i \left\|\| z _ {m,i} - \text{sg}[e _ {m,i}] \right\|\|^2$, respectively.

---

> > > > ### Comment · Reviewer_t6CB · 2024-08-13
> > > >
> > > > Thanks for these clarifications!

---

### Official Review · Reviewer_ZPqL · 2024-07-13

**Soundness:** 4
**Presentation:** 3
**Contribution:** 3
**Rating:** 6
**Confidence:** 4

**Summary:**

The authors of this paper establish the identifiability of latent state-transition processes in reinforcement learning (RL) and propose a practical method for learning these processes from observed state-action trajectories. The focus is on personalized reinforcement learning (RL), where different individuals may exhibit different state-transition processes influenced by latent individual-specific factors. The study introduces the concept of a personalized Markov decision process (iMDP) and provides theoretical guarantees for learning state-transition processes with latent factors. Experiments on multiple datasets demonstrate the effectiveness of the method in inferring these factors and learning personalized strategies.

**Strengths:**

1. **Innovation and Importance**: The paper addresses a significant challenge in RL—identifying latent state-transition processes influenced by individual-specific factors. This is crucial for optimizing personalized strategies in fields like healthcare and education.
2. **Theoretical Contribution**: The authors provide theoretical guarantees for the identifiability of latent factors, which is a significant contribution to the field of personalized RL.
3. **Experimental Validation**: The method has been experimentally validated on multiple datasets, showcasing its effectiveness in practical applications.
4. **Modular Framework**: The proposed iMDP framework is modular, allowing for independent study and improvement of each component, which is beneficial for further research and practical applications.

**Weaknesses:**

1. **Broader Experimental Scope**: Although the experiments cover multiple datasets, including more diverse environments to test the generality of the findings would be valuable, such as testing in more complex high-dimensional RL environments.
2. **Comparison with Other Methods**: A more comprehensive comparison with other state-of-the-art methods in meta-RL and multi-task reinforcement learning could strengthen the validation of the proposed method.
3. **Discussion in Appendix**: Adding a discussion in the appendix about the differences and connections between iMDP and meta-RL, context-based RL, and multi-task reinforcement learning would be beneficial.

**Questions:**

1. **Consideration of Transformer as Encoder**: In Section 5.1, "temporal dependency from the sequential observations," the use of Transformers as the primary encoder was not considered. Supplementing with relevant experimental comparisons would provide clearer insights for readers.

**Limitations:**

The paper explicitly discusses its limitations.

---

> ### Author Rebuttal · Authors · 2024-08-07
>
> **Q1: Broader experimental scope:** Thank you very much for your suggestion. In the updated manuscript, we validated our algorithm in the AhnChemoEnv in DTRGym [1] and inventory management tasks [2]. AhnChemoEnv is designed to simulate cancer treatment through chemotherapy, allowing realistic modeling of tumor growth and response to treatment. Inventory management is an important real-world problem that aims to keep inventories of goods at optimal levels to minimize inventory costs while maximizing revenue from demand fulfillment. We tested the performance of our algorithm on the inventory with state dimensions of 50, 100, and 200. The experimental results (see Figure 1 and Figure 3 in REBUTTAL.pdf) show that our framework outperforms other algorithms in terms of initial reward and final reward. Detailed analyses are provided in Q2.
>
> [1] Luo, Zhiyao, et al. "DTR-Bench: An in silico Environment and Benchmark Platform for Reinforcement Learning Based Dynamic Treatment Regime." arXiv preprint arXiv:2405.18610 (2024).
>
> [2] Sun, Yuewen, et al. "ACAMDA: Improving Data Efficiency in Reinforcement Learning Through Guided Counterfactual Data Augmentation." Proceedings of the AAAI Conference on Artificial Intelligence. Vol. 38. No. 14. 2024.
>
> **Q2: Comparison with other methods:** We appreciate your suggestion. In the updated manuscript, we have added more benchmarks and further compared our algorithm against (1) meta gradient RL, (2) multitask RL, (3) policy distillation, and (4) non-policy adaptation. Additionally, we have included two evaluation metrics: (1) initial reward, which measures the initial performance benefited from policy adaptation, and (2) final reward, which measures the performance after the full training process. Please refer to REBUTTAL.pdf for the detailed results.
>
> The comparisons suggest that our method achieves the highest initial and final rewards compared to the benchmarks. Specifically, it shows a significant jump-start compared to non-policy adaptation, validating the effectiveness of our adaptation approach. The meta-gradient method optimizes the hyperparameters of the learning algorithm by calculating the gradient of the learning process, allowing rapid adaptation to new tasks as they change. However, due to the continuous adjustment of learning strategies during training, it converges more slowly and the adaptation effect is less significant compared to our algorithm.
>
> Multitask RL improves learning efficiency by sharing model strategies across different tasks. This requires first training policies on multiple tasks, which can be time-consuming (and even risky) during exploration. Moreover, identifying which new task corresponds to a previously trained task can be challenging. Our algorithm addresses this by estimating directly using $\kappa$ without requiring prior knowledge.
>
> Policy distillation transfers the knowledge of already trained teacher models to a student model, allowing the student to perform well across multiple tasks. However, this approach highly relies on the performance of the teacher models; insufficiently trained teacher models can negatively impact the final performance. Our algorithm does not depend on the source policy performance; subsequent policy optimization is based on the new environment, leading to better final performance.
>
> **Q3: Discussion in Appendix:**  In the updated manuscript, we have added a section discussing the differences and connections between iMDP and meta-RL, context-based RL, and multi-task reinforcement learning, summarized below.
>
> - **Meta-learning** trains a learning model on a variety of tasks so that it can efficiently apply what it has learned to new tasks. Unlike our method, it does not assume a time-invariant latent factor, has no guarantee of identifiability, and does not provide a clear clue of adaptation. iMDP captures how an individual's belonging to a certain group affects their interactions within an environment, allowing for individualized policy adaptation. Moreover, iMDP provides a guarantee of identifiability and develops a corresponding estimation framework that potentially offers better interpretability.
>
> - **Contextual MDPs** consider the general contextual influence on transition probabilities and rewards. However, the context variables are assumed to be partially observable and do not guarantee the identifiability of the context variables. In our case, when the latent factors are finite, our method guarantees group-wise identifiability even when the transition processes are nonparametric. In the cases of infinite latent factors, identification could be achieved under proper assumptions.
>
> - **Multi-task RL** involves learning policies for a variety of tasks simultaneously. The goal of the agent is to perform well on all these tasks, which may have similar or different objectives. It often involves sharing information between tasks to improve learning efficiency and policy performance. Instead of focusing on policy optimization for all tasks, iMDP identifies latent individual-specific factors that implicitly influence the decision-making process. These factors indicate the unique properties of each individual, providing explanatory clues for policy adaptation.
>
> **Q4: Consideration of transformer as encoder:** We appreciate your insightful suggestion. In the revision, we have incorporated transformers into our framework to perform a comparative analysis against the models currently in use, and the results are reported in Figure (c) in REBUTTAL.pdf. Although both frameworks can achieve identifiability, the Transformer encoder can achieve faster convergence compared with our previous framework.

---

### Official Review · Reviewer_cEn6 · 2024-07-15

**Soundness:** 3
**Presentation:** 3
**Contribution:** 3
**Rating:** 7
**Confidence:** 4

**Summary:**

The paper titled "Identifying Latent State-Transition Processes for Individualized Reinforcement Learning" addresses the challenge of optimizing individualized reinforcement learning (RL) policies by focusing on latent individual-specific factors that influence state transitions. This is particularly significant in domains like healthcare and education, where individual differences can causally affect outcomes.

The primary contribution of the paper is the establishment of the identifiability of latent factors that drive individualized state-transition processes. The authors propose a novel framework called Individualized Markov Decision Processes (iMDPs) that incorporates these latent factors into the RL framework. The key advancements include:

1. The paper provides theoretical guarantees for the identifiability of latent factors under both finite and infinite conditions, making it possible to distinguish different underlying components in state transitions.

2. The authors develop a generative-based method to effectively estimate these latent individual-specific factors from observed state-action trajectories. This method employs a variational autoencoder with a vector quantization layer to discretize the latent space and ensure accurate factor estimation.

3. The proposed method is empirically validated across various datasets, demonstrating its effectiveness in identifying latent state-transition processes and improving the learning of individualized policies.

4. The paper outlines a two-stage approach for policy learning, where the estimated latent factors are used to tailor RL policies for individuals, thereby enhancing the policy’s adaptability to different environments and individuals.

**Strengths:**

- The introduction of the Individualized Markov Decision Processes (iMDPs) framework is a novel approach to integrating latent individual-specific factors into reinforcement learning (RL). This approach goes beyond existing models by considering the fixed influence of these latent factors on state transitions, addressing a significant gap in the current literature.
   - The paper establishes theoretical guarantees for the identifiability of these latent factors under both finite and infinite conditions.
   - The use of variational autoencoders (VAEs) with vector quantization to discretize the latent space and estimate the latent factors is not surprisingly a straightforward application of these techniques in the context of RL.
   - The experimental validation is thorough, with the proposed method tested on multiple datasets, including synthetic data and real-world scenarios like the Persuasion For Good corpus. The experiments are well-designed to demonstrate the effectiveness of the method in different settings, showcasing its robustness and versatility.

**Weaknesses:**

**Assumptions on Latent Factors**
The paper assumes that the latent individual-specific factors are time-invariant, which might not hold in all real-world scenarios. For instance, in healthcare, a patient’s condition can change over time, affecting the state-transition processes. Addressing this limitation could involve extending the framework to account for time-varying latent factors or providing a discussion on how to handle such scenarios.

**Limited Real-World Validation**
While the empirical validation includes synthetic datasets and a real-world dataset (Persuasion For Good corpus), the application domains are somewhat limited. The healthcare and education examples are discussed theoretically but not empirically validated with real-world data. To strengthen the paper, the authors could include experiments using real-world healthcare or education datasets to demonstrate the practical utility and robustness of their method in these critical applications. A possible healthcare platform is [1], where there are 4 environments with customizable PK/PD variation setups.

**Policy Adaptation for New Individuals**
The approach for policy adaptation for new individuals involves initializing the policy based on the estimated group factor and fine-tuning it with new interactions. However, the paper lacks detailed evaluation metrics or benchmarks to assess the effectiveness and efficiency of this adaptation process. Including more quantitative analysis and comparisons with existing methods for policy, adaptation would enhance the understanding of the benefits and limitations of the proposed approach.

### Actionable Suggestions
1. add a discussion of time-varying latent factors in your limitation section
2. Incorporate real-world experiments in healthcare or education to validate the practical utility of the method.
3. Provide detailed evaluation metrics and benchmarks for the policy adaptation process.


References
[1] Luo, Zhiyao, et al. "DTR-Bench: An in silico Environment and Benchmark Platform for Reinforcement Learning Based Dynamic Treatment Regime." arXiv preprint arXiv:2405.18610 (2024).

**Questions:**

How effective is the policy adaptation process for new individuals compared to existing methods? Can the authors provide quantitative benchmarks or metrics to evaluate this process?

How does the variability in initial states across individuals impact the performance and identifiability of the latent factors? Are there specific strategies to handle high variability in initial states?

**Limitations:**

Privacy Concerns: The individualized nature of the proposed method implies that it relies on detailed personal data. This raises potential privacy concerns, especially in sensitive domains like healthcare and education. The authors should discuss how they plan to handle privacy issues, possibly suggesting the use of privacy-preserving techniques such as differential privacy.

Robustness and Reliability: Another potential impact is related to the robustness and reliability of the individualized policies. In critical applications, such as healthcare, incorrect policy recommendations could have serious consequences. The authors should address its limitations from this angle.

---

> ### Author Rebuttal · Authors · 2024-08-07
>
> **Q1: Discussion of time-varying latent factors:** Thank you for pointing out this interesting scenario. It is indeed possible for the latent individual-specific factor to be time-variant in the considered problem. We believe our framework can be extended to handle time-varying cases, although establishing the theoretical identifiability is highly non-trivial. For instance, if we directly allow these latent factors to be time-varying, it seems hopeless to recover the latent variables, since each individual may have a specific latent factor. Therefore, further constraints would be needed. For instance, the theoretical identifiability might benefit from the constraint that the total number of possible values of the latent factors over time and across different individuals is finite. We hope this important extension can be achieved by researchers in the field by leveraging our framework and further considering the idea in [1].
>
> [1] Hu, Yingyao, and Matthew Shum. "Nonparametric identification of dynamic models with unobserved state variables." Journal of Econometrics 171.1 (2012): 32-44.
>
> **Q2: Additional real-world experiments in healthcare:** Thank you very much for your suggestion and for providing the healthcare platform. In the updated manuscript, we validated our algorithm in the AhnChemoEnv environment in DTRGym [2] and created different groups with PK/PD variation. The experimental results (see Figure 1 in REBUTTAL.pdf) show that our framework outperforms other algorithms in terms of initial & final reward. Detailed analyses are provided in Q3.
>
> [2] Luo, Zhiyao, et al. "DTR-Bench: An in silico Environment and Benchmark Platform for Reinforcement Learning Based Dynamic Treatment Regime." arXiv preprint arXiv:2405.18610 (2024).
>
> **Q3: Evaluation and discussion on policy adaptation process:** We appreciate your suggestion. In the updated manuscript, we have added more baselines and further compared our algorithm against (1) meta gradient RL, (2) multitask RL, (3) policy distillation, and (4) non-policy adaptation. Additionally, we have included two evaluation metrics: (1) initial reward, which measures the initial performance benefited from policy adaptation, and (2) final reward, which measures the performance after the full training process. Please refer to REBUTTAL.pdf for the detailed results.
>
> Our method achieves the highest initial and final rewards compared to the baselines. Specifically, it shows a significant jump-start compared to non-policy adaptation, validating the effectiveness of our adaptation approach. The meta-gradient method optimizes the hyperparameters of the learning algorithm by calculating the gradient of the learning process, allowing rapid adaptation to new tasks as they change. However, due to the continuous adjustment of learning strategies during training, it converges more slowly and the adaptation effect is less significant compared to our algorithm.
>
> Multitask RL improves learning efficiency by sharing model strategies across different tasks. This requires first training policies on multiple tasks, which can be time-consuming (and even risky) during exploration. Moreover, identifying which new task corresponds to a previously trained task can be challenging. Our algorithm addresses this by estimating directly using $\kappa$ without requiring prior knowledge.
>
> Policy distillation transfers the knowledge of already trained teacher models to a student model, allowing the student to perform well across multiple tasks. However, this approach highly relies on the performance of the teacher models; insufficiently trained teacher models can negatively impact the final performance. Our algorithm does not depend on the source policy performance; subsequent policy optimization is based on the new environment, leading to better final performance.
>
> **Q5: Discussion on variability in initial states:** This is a great question! We can still establish the identifiability of the latent variable for each trajectory provided by each user. However, since each trajectory has a finite (usually pretty small) length, the estimated values might be noisy. The second step is to merge latent variables that should have the same value. A straightforward method is as follows: we first estimate the value of the latent variable for each individual and then see how they can be clustered together. We check whether they should be classified into the same or different groups by conducting clustering.
>
> A more principled way to address this issue has yet to be developed, and we have conducted additional experiments to verify the feasibility and report the results in Figure (a-b) in REBUTTAL.pdf. The initial state distributions are defined with two types: normal and uniform. For the normal distribution, the means are set to [0, 1, 1] and the standard deviations are set to [1, 2, 1], respectively. For the uniform distribution, the range for each dimension is defined with lower bounds [0, -1, 1] and upper bounds [1, 1, 1.5]. As you can see from the experiment, although the initial state has a high variability, finally the estimated values of the latent factors corresponding to the 200 individuals are highly classified and can be divided into 4 groups.
>
> **Q6: Impact on privacy issues \& robustness and reliability:** In the updated manuscript, we have included a separate limitation section to discuss how we address these limitations.
>
> - Potential privacy risks: We could use de-identifying techniques and remove direct identifiers such as names and zip codes, apply masking techniques such as data perturbation, and use pseudonymization to replace private identifiers with artificial ones. Incorporating differential privacy techniques is also interesting and we will leave it as future work.
>
> - Robustness and reliability: We could provide an assumption checklist to help users determine whether our work is applicable to their specific scenarios, thereby avoiding misuse and improving reliability.

---

> > ### Comment · Reviewer_cEn6 · 2024-08-12
> >
> > Thank you for the response. The authors have addressed most of my major concerns.
> > I have no further questions, and I am happy to raise the score to 7.

---

> > > ### Author Response · Authors · 2024-08-12
> > >
> > > Dear Reviewer cEn6,
> > >
> > > Thank you so much for your time and feedback. It means a lot to us. We are so happy that most of your major concerns were properly addressed.
> > >
> > > Wish you all the best,
> > >
> > > Authors of submission 12027

---

### Author Rebuttal · Authors · 2024-08-07

We thank all reviewers for their time dedicated to reviewing the paper and valuable comments. We have revised the manuscript accordingly as described below. Concerns about the experiments are addressed collectively in the REBUTTAL.pdf. Your further feedback, if any, would be appreciated.

---

### Author Response · Authors · 2024-08-12
**Thanks to All Reviewers. Are There Any Other Concerns?**

Dear Reviewers,

Thank you very much for your time and effort in reviewing our manuscript. We have tried our best to address all your concerns and have revised our paper accordingly.

As the discussion deadline approaches, please let us know if you have any further concerns and we will be happy to address them.

Sincerely,

Authors of Submission #12027

---

### Decision · Program_Chairs · 2024-09-25

**Decision:**

Accept (poster)

**Comment:**

This paper demonstrates the identifiability of latent factors and introduces a practical method for learning these processes from observed state-action trajectories. Experiments across a few [somehow similar] domains confirm that the method can effectively identify latent state-transition processes and help developing individualized RL policies. While in general the paper's presentation is good, it heavily relies on theoretical results which are provided in the appendix. As such supplementary material is not reviewed here, I would strongly recommend that the authors double-check all the results, proofs, and examples in the appendix before camera-ready version.